# LARGE LANGUAGE MODELS ENHANCE GRAPH LEARNING WITHOUT GRAPH SERIALIZATION

## ABSTRACT

We introduce GINAT, a framework that augments graph neural networks (GNNs) with global graph descriptions encoded by large language models (LLMs). While prior work on text-attributed graphs (TAGs) integrates text features tied to nodes or edges, our approach leverages external prompt datasets that describe graph semantics independently of the downstream task. These prompts are embedded by a frozen LLM and injected into the message-passing process via cross-attention, allowing GNNs to incorporate global contextual information during local computations. We provide a theoretical analysis showing that GINAT provably increases the expressive capacity of GNNs while preserving permutation equivariance.

Extensive experiments across diverse graph benchmarks demonstrate consistent improvements over strong GNN baselines, with performance gains of up to 10%. Ablation studies further confirm that these improvements stem from the semantic content of the prompts rather than from random noise or model size. Together, our findings highlight LLM-embedded prompts as a principled and effective new modality for enhancing graph learning.

Our code is available at `https://anonymous.4open.science/r/llm4gfm-3FBE/`

## 1 INTRODUCTION

*What do graphs and natural language have in common?* At first glance, they appear to be fundamentally different. Graphs offer a formal framework for decomposing objects, concepts, and complex systems into smaller components and the relationships among them. They enable precise, unambiguous representations that support mathematical analysis and computational processing. Natural language, in contrast, provides a flexible medium for high-level discourse, abstracting away fine details while enriching communication with context, nuance, and cross-domain associations (Gentner & Asmuth, 2019; Williams et al., 2017; Korta & Perry, 2024). In this work, we regard graphs and natural language as *complementary* representational paradigms, linked by a fundamental commonality: both constitute human-engineered abstractions devised to model, organize, and interpret the complexity of the world.

The past decade has seen the rise of deep learning models that broke ground in natural language processing (NLP) and graph processing. Graph neural networks (GNNs) have made tremendous contributions in drug design and related biochemical tasks (Jumper et al., 2021; Vignac et al., 2023), recommendation systems (Wang et al., 2019), among other fields. LLMs de-facto solved most of the NLP tasks, and are widely believed to be the bringers of artificial intelligence and artificial generative intelligence (Qin et al., 2024; Mumuni & Mumuni, 2025).

Encouraged by the success of foundation models in other modalities such as language (Mumuni & Mumuni, 2025) and vision (Awais et al., 2025), the graph learning community has made efforts towards equivalent models for the graph modality. Towards this goal, multiple approaches were proposed; examples include pretraining for GNNs (Hou et al., 2022), mixture-of-experts (Zhao et al., 2024), and LLM-based inference (Liu et al., 2024).

Motivated by these developments, we adopt a novel perspective on the relationship between graphs and natural language. Our approach stems from the following fundamental hypothesis: *Graph inference would benefit from global information about the input graph*. As stated earlier, graph

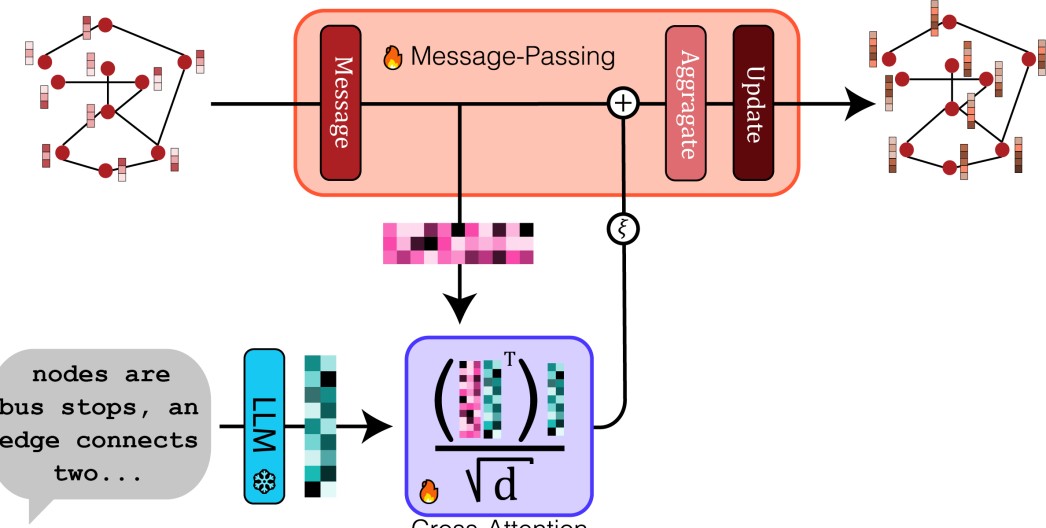

Figure 1: Illustration of GINAT. A graph with initial features is associated with a prompt that describes it. We enrich traditional message passing by cross-attention between graph messages with a representation of its natural language description, generated by a frozen, pretrained LLM. The output of the cross-attention operation is then scaled by a learned parameter $\xi$ and summed with the original messages.

representation of data and textual description complement each other. For example, a cycle in a molecular graph has a different meaning than a cycle in a transportation graph. When humans reason about a graph, they have *prior knowledge* about what it represents, and how to give meaning to substructures. We argue that providing graph models with such prior knowledge is essential towards graph foundation models.

**Our Approach.** In this work, we augment graphs with textual descriptions. In particular, our approach consists of two parts: data and technique. Given a graph of a certain nature (for example: molecular data, citation networks, social networks and others), we associate the graph with a prompt that describes it. Such a description can be about the graph as a whole (e.g., `this graph represents a molecule`) or as a sum of its components (e.g., `nodes are papers and edges indicate citation`). In Figure 3 and Appendix H, we provide concrete examples of such prompts. We note that the prompts we use do not contain information on any downstream task.

Therefore, we call our approach GINAT: **G**raph **I**nference by **N**atural-language **A**ugmentation of **T**opology. Our framework augments message-passing layers with prompt embeddings encoded by a pre-trained LLM, followed by a cross-attention processing step with messages passed on the graph's edges by said layers. Figure 1 illustrates the approach, and Section 3 provides implementation details. The benefits are twofold: the prompts are enriched by the prior knowledge learned by the LLM, and the GNN receives explicit graph information.

Our contributions are summarized as follows:

- In Section 3, we propose GINAT for graph inference using natural-language data external to the dataset. We also provide the datasets of graph-descriptive prompts.

- In Section 4, we provide extensive theoretical analysis of GINAT, and prove that it improves the capacity of the GNN while maintaining permutation equivariance.

- In Section 5, we establish the potential of the approach. We show improved results on a variety of datasets and using different backbone GNNs.

## 2 RELATED WORK

In this section, we present ongoing lines of research related to our work.

### 2.1 TEXT-ATTRIBUTED GRAPHS

Text-attributed graphs (TAGs) combine graph structure with textual node features and appear in many domains, from citation networks to e-commerce. Several recent works advance TAG research. Yan et al. (2023a) introduces a comprehensive benchmark with new datasets, tasks, and systematic evaluation. Yan et al. (2023b) propose GLEM, a scalable framework for large TAGs using variational inference and message passing. Zhang et al. (2024) present the first benchmark for DyTAGs (dynamic TAGs), curating large-scale time-evolving graphs and defining four tasks. Yan et al. (2024) develop an LLM-to-LM Interpreter that distils explanations from LLMs to enrich TAG representations.

While this progress inspired our work, our approach differs fundamentally. TAGs integrate text as node features, whereas we leverage external text that is independent of the target dataset and agnostic to the downstream task, describing instead the semantics of the graph and its sub-entities.

### 2.2 LARGE LANGUAGE MODELS AS ENHANCERS FOR GRAPH NEURAL NETWORKS

LLMs offer rich semantic priors that can complement the structural focus of GNNs, making them especially relevant for foundation-style graph models. Prior work highlights several roles for LLMs: as *knowledge injectors*, filling gaps in incomplete graphs and improving link prediction through alignment with textual embeddings Li et al. (2023b); Chen et al. (2024a); Tian et al. (2024); as sources of *self-supervised signals*, generating semantic objectives that enrich pretraining beyond structural tasks You et al. (2024); Li et al. (2023b); and as *interpretable enhancers*, producing natural-language rationales that improve transparency and support human-in-the-loop reasoning Gu et al. (2024).

### 2.3 GRAPH NEURAL NETWORKS AS ENHANCERS FOR LARGE LANGUAGE MODELS

Recent work explores how GNNs enrich LLMs with structured relational reasoning. While LLMs excel at contextual language understanding, they remain limited in explicitly modeling entities and relations. GNNs, with their inductive bias for graph data, have been integrated into LLM-based systems in several ways. One line of work uses knowledge graph augmentation, where GNNs encode entities and relations into embeddings fused with LLM representations. Examples include LinkBERT, which leverages inter-document links during pretraining Yasunaga et al. (2022), and methods that inject graph-based entity embeddings to improve factual QA and reasoning Zhang et al. (2023).

A second line investigates hybrid GNN–LLM architectures. Molecular graph encoders combined with LLMs enhance drug discovery tasks Wang et al. (2022), while abstract syntax tree encodings integrated with Transformers improve code reasoning Hellendoorn et al. (2020). More recently, graph-guided prompting has emerged, where GNNs extract relational substructures (e.g., multi-hop paths, neighborhoods) and feed them to LLMs as structured prompts, enabling multi-hop reasoning without modifying the architecture Chen et al. (2024b).

Challenges remain in *scalability*, due to high communication and memory costs in large graphs Shao et al. (2022); Huang et al. (2024); Liu et al. (2023); in *semantic alignment*, bridging graph and token embeddings Wang et al. (2024); Li et al. (2023b); and in *evaluation*, with new benchmarks such as GLBench and LLMNodeBed beginning to standardize comparisons Zhou et al. (2024); Chen et al. (2025).

### 2.4 CROSS-ATTENTION FOR GRAPH LEARNING

Cross-attention is used for reasoning that jointly leverages symbolic graph structure and natural language priors, with promising applications in knowledge graph completion (Yao et al., 2022), scientific discovery (Yang et al., 2023), and multi-hop reasoning tasks (Chen et al., 2023). Ongoing challenges include balancing modality dominance, designing scalable cross-attention schemes for large graphs, and ensuring semantic alignment between embedding spaces.

## 3 METHOD

In this section, we present GINAT. Our approach extends the classical message-passing paradigm in graph learning. Message-passing layers are conventionally divided into three abstract components: *message*, *aggregate*, and *update* (see Appendix F for more information). In GINAT, we redefine the message computation step by incorporating semantic information derived from a pre-trained, but not finetuned, LLM. Each graph is associated with a prompt that describes *the nature* of the graph, that is, what information it represents. The prompt is processed by an LLM, and the output embeddings of its last layer are employed to enrich the original messages before aggregation, thereby introducing contextualized knowledge in the graph learning process. This idea is illustrated in Figure 1, and we provide examples for the prompts used in Figure 3.

**GINAT: Formal Introduction.** Let us consider a message-passing layer defined by the $message(\cdot), aggregate(\cdot), update(\cdot)$ functions. In GINAT, we make no changes to those methods, but we change the data flow from the $message(\cdot)$ to $aggregate(\cdot)$. Specifically, let $G = (V, E)$ be a graph with $|V|$ nodes and $|E|$ edges, with $\mathbf{H}^{(l)} \in \mathbb{R}^{|V| \times d_{l-1}}$ being node embeddings at layer $l$. Finally, we denote by $\mathcal{H}_T \in \mathbb{R}^{L \times d_T}$ the LLM-generated embedding of the prompt associated with the graph, and a learnable linear projection matrix $\mathbf{W} \in \mathbb{R}^{d_l \times d_T}$. The output $\mathbf{H}^{(l+1)}$, that is, the augmented message-passing layer, is then obtained via the following sequence of steps:

$$\mathbb{R}^{|L| \times d_l} \ni \quad \tilde{\mathcal{H}}_T = \text{LayerNorm}(\mathcal{H}_T) \mathbf{W}^T, \tag{1}$$

$$\mathbb{R}^{|E| \times d_l} \ni \quad \mathbf{M}^{(l)} = message\left(\mathbf{H}^{(l)}, E\right), \tag{2}$$

$$\mathbb{R}^{|E| \times d_l} \ni \quad \tilde{\mathbf{M}}^{(l)} = \mathbf{M}^{(l)} + \xi \cdot \text{CrossAttention}\left(\tilde{\mathbf{M}}^{(l)}, \tilde{\mathcal{H}}_T\right), \tag{3}$$

$$\mathbb{R}^{|V| \times d_l} \ni \mathbf{H}^{(l+1)} = update\left(\mathbf{H}^{(l)}, aggregate\left(\tilde{\mathbf{M}}^{(l)}\right)\right). \tag{4}$$

Figure 1 illustrates this process in a more intuitive manner. The CrossAttention term allows passed messages to selectively attend over contextualized prompt embeddings $\mathcal{H}_T$. The resulting formulation unifies structural and semantic information, offering a principled way of integrating pretrained language representations into graph message passing. Importantly, when the cross-attention weights are set to zero, the formulation reduces to the standard MPNN, implying that our method is at least as expressive as the MPNN we extend, in the sense of the WL test.

**Scaling the Cross-Attention Output.** $\xi$ in Equation (3) is a learned scaler, that serves two purposes. First, it scales the cross-attention output. This allows the model to learn the appropriate scale of the attention mechanism compared to the MPNN's output. This allows stable training and prevents vanishing gradients due to scale mismatch between the components. Second, it provides **explainability**. The value of $\xi$ implies how important the model finds the information carried in the prompt embedding to be. Furthermore, it allows the model to lean $\xi = 0$, implying the added information isn't relevant to the task.

**Graph Level Aggregation.** For graph-level classification, GINAT employs a virtual node, connected to all nodes in the target graph. After the forward pass, this node's output representation is treated as the graph's latent features.

### 3.1 THE CONTRIBUTION OF LLMS IN GINAT

By leveraging the vast knowledge encoded in LLMs, trained on massive corpora of text data, our method goes beyond the standard setting TAGs. Instead, it provides a more general framework in which external semantic priors complement structural message passing, thereby enhancing the representational capacity of even simple GNN architectures. The effectiveness of this integration is demonstrated through empirical results presented in Section 5.

**Choice of LLM.** LLMs can be proprietary or open-source, and used in pre-trained, fine-tuned, or instruction-tuned configurations. In this work, we use open-source models in their pre-trained form, since the LLM serves only as an encoder, and this provides direct access to its prior knowledge,

unaffected by alignment or tuning. While fine-tuning within our framework is a promising direction, it is resource-intensive; we focus here on testing the hypothesis that textual graph descriptions encoded by an LLM enhance inference.

# 4 THEORETICAL PROPERTIES OF GINAT

In this section, we provide an analysis of GINAT from model design and theoretical perspectives. We begin by presenting the relevance of cross-attention to our goal, continue by formally establishing the enhanced representation capabilities of our approach, and conclude by proving that we maintain permutation equivariance.

Message passing is the dominant framework for modern GNNs, formalizing relational information exchange between nodes. At each iteration $t+1$, a node updates its representation by aggregating messages from neighbors' features at iteration $t$ via permutation-invariant operators such as sum, mean, or attention (Kipf & Welling, 2017; Hamilton et al., 2017; Veličković et al., 2018). This captures local dependencies while preserving permutation equivariance (Appendix B). Stacking multiple layers allows nodes to integrate information from larger neighborhoods, enabling GNNs to approximate diverse graph functions. However, their expressivity is limited by the 1-WL test (Xu et al., 2019), motivating extensions such as higher-order architectures (Morris et al., 2019), positional encodings (Rampášek et al., 2022; Eliasof et al., 2023), and random feature initializations Abboud et al., 2020.ions Abboud et al., 2020.

We follow these trends and propose another method to extend the expressive power of an MPNN. We equip the MPNN with a cross-attention mechanism (described in detail in Appendix E), which learns to extract relevant information from global, external feature vectors (e.g., embedded prompts). This is used to enrich the messages passed by the MPNN and extend its capacity.

## 4.1 ENHANCED GRAPH REPRESENTATION CAPABILITIES

Our approach can successfully distinguish between graph structures that standard MPNNs fail to differentiate. It is well established that MPNNs are at most as expressive as the 1-WL isomorphism test Xu et al. (2019). However, the 1-WL test is known to be incapable of distinguishing between certain families of graphs, such as $k$-regular graphs (where every node has degree $k$). A classical example is the cycle graph $C_6$ and the disjoint union of two triangles ($C_3 \cup C_3$), illustrated in Figure 2. The former consists of a single 6-node cycle, while the latter consists of two disconnected 3-cycles. For the interested reader, we provide a proof for the failure of MPNNs to separate these two

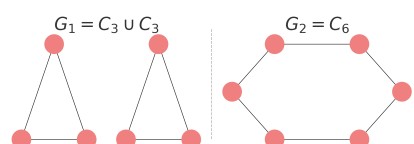

Figure 2: Illustrations of $G_1, G_2$ as described in Section 4.1. These non-isomorphic graphs are indistinguishable by standard MPNNs.

different graphs in Lemma 3 in Appendix A. To establish the benefits of our GINAT, we prove the following lemma in Appendix A:

**Lemma 1** (GINAT Achieves Separation Given Informative Global Features). *Let $G_1$ be the graph comprised of two disconnected triangles and $G_2$ the graph containing a 6-cycle. Let both graphs be initialized such that every node has an identical feature vector $x$. With each graph, we associate a global feature vector $g_1, g_2$, respectively. Redefine the message sent to each node during message passing as*

$$m_{vu}^{(l)} = message\left(h_u^{(l-1)}\right), \ u \in N(v), \tag{5}$$

$$m_{vu}'^{(l)} = m_{vu}^{(l)} + \text{CrossAttention}\left(m_{vu}^{(l)}, g\right) \tag{6}$$

*where $h_v^{(l)} = update\left(h_v^{(l-1)}, aggregate\left(\left\{\left(message\left(h_u^{(l-1)}\right)\right) \mid u \in N(v)\right\}\right)\right)$. For a 2-layer MPNN, there exists a choice of $g_1 \neq g_2$ such that the final readout of both graphs is distinct, $H_{G_1} \neq H_{G_2}$, and there exists a choice of $g_3 \neq g_4$ (which are not linearly dependent) such that the final readout of both graphs is identical, $H_{G_1} = H_{G_2}$.*

> [ "Molecular structure where atoms are nodes and
>   edges are their connecting bonds.",
>
>  "A graph where nodes are atoms and edges show their
>  covalent connections.",
>
>  "Each node signifies an atom, and each edge
>  signifies a chemical bond.", ... ]

Figure 3: Examples of prompts, used to describe molecular graphs, such as $OGBG - MOLHIV$ and ZINC. The entire prompts dataset is provide in Appendix H.

Lemma 1 establishes that an MPNN equipped with GINAT can separate $G_1$ from $G_2$ given extra feature vectors that provide *relevant information* on the graphs, or a specific task. Furthermore, it follows from the Lemma proof that when the given extra feature vectors are *not informative*, our approach will not separate the graphs. This is a *desired property*: augmenting two graphs with different feature vectors in order to obtain distinct representations is trivial, while GINAT has the ability to augment the input graph data in an informed and adept way.

**On the Benefit of LLM-embedded Prompts.** LLMs provide the ability to meaningfully represent text, while also using the rich prior they learn during pretraining. This could yield a representation of text that not only captures the meaning, but also provides context and additional information not present in the raw text. This capability makes LLMs the perfect candidates to generate informative extra features, as the results in Section 5 show.

**Permutation Equivariance in GINAT.** Permutation equivariance is a core property of GNNs and graph learning techniques. While it is well-established that cross-attention is permutation equivariant, we provide a concrete proof for this statement in Proposition 5. Proposition 2 establishes that our approach maintains permutation equivariance of MPNNs with a complete proof given in Appendix B:

**Proposition 2** (GINAT Preserves Permutation Equivariance). *Standard GNNs are permutation equivariant: permuting the input node order results in the same permutation applied to the output embeddings. This property is preserved if the message function incorporates cross-attention.*

*Formally, consider a message passing layer of the form*

$$\tilde{h}_v^{(t+1)} = update\left(h_v^{(t)},\ aggregate\left(\{m_{uv}^{(t)} : u \in \mathcal{N}(v)\}\right),\ \mathrm{CrossAttention}\left(h_v^{(t)}, H_T\right)\right), \quad (7)$$

*where $h_v^{(t)}$ is the embedding of node $v$ at layer $t$, $\mathcal{N}(v)$ denotes its neighborhood, $m_{uv}^{(t)}$ is the message from node $u$ to $v$, $update$ is a permutation-invariant aggregation function, $\mathrm{CrossAttention}$ is a cross-attention operator as defined in Appendix E with $h_v^{(t)}$ as query and $H_T$ as keys/values, and $update$ is the node-wise update function. Then the resulting GNN layer is permutation equivariant with respect to the nodes of the input graph.*

## 5   EXPERIMENTS

To demonstrate the utility of GINAT and better understand its impact, we designed experiments structured around the following guiding questions:

(Q1) Can GINAT provide evidence that our core hypothesis is true, and GNNs can benefit from a global description of the processed graph?

(Q2) Are the gains observed with GINAT attributed to the informative embeddings produced by the LLM, rather than to indirect computational effects?

(Q3) How does GINAT contribute to enhancing the overall capability of GNNs?

Table 1: Graph Performance on Molecular Datasets. Some of the datasets contain edge features; We used those features in GAT, GATv2, GIN[E] that support them by definition. GINAT results are highlighted in red. Best results on for each GNN-dataset pair are in **bold**.

| Model ↓ Metric → | MUTAG Acc. ↑ | ENZYMES Acc. ↑ | PROTEINS Acc. ↑ | NCI1 Acc. ↑ | NCI109 Acc. ↑ | MOLHIV ROC-AUC ↑ | ZINC MAE ↓ |
|---|---|---|---|---|---|---|---|
| GCN | 0.633 ±0.17 | 0.344 ±0.10 | 0.685 ±0.02 | 0.737 ±0.02 | 0.748 ±0.03 | 0.769 ±0.01 | 0.392 ±0.01 |
| | **0.733** ±0.12 | **0.533** ±0.04 | **0.744** ±0.03 | **0.763** ±0.02 | **0.791** ±0.01 | **0.785** ±0.02 | **0.391** ±0.08 |
| GraphSAGE | 0.667 ±0.10 | 0.250 ±0.02 | 0.732 ±0.02 | 0.750 ±0.03 | 0.750 ±0.01 | 0.728 ±0.01 | 0.517 ±0.02 |
| | **0.733** ±0.10 | **0.633** ±0.00 | **0.759** ±0.02 | **0.763** ±0.03 | **0.791** ±0.01 | **0.770** ±0.02 | **0.502** ±0.00 |
| GAT | 0.667 ±0.08 | 0.394 ±0.03 | 0.670 ±0.04 | 0.702 ±0.07 | 0.683 ±0.05 | 0.721 ±0.02 | 0.402 ±0.02 |
| | **0.767** ±0.06 | **0.506** ±0.08 | **0.720** ±0.04 | **0.783** ±0.01 | **0.775** ±0.02 | **0.748** ±0.01 | **0.344** ±0.02 |
| GATv2 | 0.667 ±0.10 | 0.339 ±0.05 | 0.673 ±0.03 | 0.738 ±0.02 | 0.718 ±0.02 | 0.712 ±0.01 | 0.359 ±0.01 |
| | **0.783** ±0.05 | **0.500** ±0.04 | **0.729** ±0.06 | **0.770** ±0.02 | **0.791** ±0.02 | **0.751** ±0.01 | **0.315** ±0.01 |
| GIN[E] | 0.650 ±0.04 | 0.672 ±0.03 | 0.747 ±0.03 | 0.807 ±0.03 | 0.821 ±0.01 | 0.753 ±0.01 | 0.233 ±0.01 |
| | **0.700** ±0.10 | **0.683** ±0.04 | **0.753** ±0.06 | **0.817** ±0.03 | **0.833** ±0.01 | **0.784** ±0.00 | **0.136** ±0.00 |
| GraphConv | 0.667 ±0.09 | 0.517 ±0.17 | 0.723 ±0.04 | 0.806 ±0.02 | 0.794 ±0.00 | 0.768 ±0.00 | 0.225 ±0.02 |
| | **0.733** ±0.15 | **0.594** ±0.03 | **0.744** ±0.04 | **0.817** ±0.03 | **0.816** ±0.01 | **0.788** ±0.01 | **0.197** ±0.00 |
| GraphGPS | 0.783 ±0.09 | 0.644 ±0.15 | 0.482 ±0.10 | 0.799 ±0.03 | 0.808 ±0.01 | 0.752 ±0.02 | 0.216 ±0.06 |
| | **0.817** ±0.09 | **0.694** ±0.03 | **0.765** ±0.04 | **0.805** ±0.04 | **0.819** ±0.01 | **0.777** ±0.02 | **0.186** ±0.01 |

## 5.1 EXPERIMENTAL SETUP

In our experiments, we integrate GINAT into widely used GNN baselines (Tables 1, 2). The LLM is frozen and used only once to embed prompts, producing fixed representations for downstream tasks. Results are averaged over three runs with distinct, predefined seeds, and for each run we trained GINAT for 100–300 epochs, reporting the test score from the best validation checkpoint. All baseline results are from our own runs under identical splits and configurations to ensure fairness. Pretrained LLMs were obtained from HuggingFace, and we use backbone architectures PyTorch Geometric implementations (Fey & Lenssen, 2019). In all experiments, we initialized the cross-attention scaler $\xi = 0.1$ (see Section 3 for more details).

**Models' Parameters Count.** Since GINAT utilizes cross-attention with LLM embeddings, whose feature dimensions are on the order of $10^3$, the layers in GINAT are larger than those in the GNN baselines typically reported in graph learning literature. To maintain a fair comparison, and to rule out the possibility that improvements arise merely from increased capacity and larger parameter count, we assign GNN baselines the same parameter budget as GINAT. Specifically, we retain architectural and configurational similarity by evaluating baselines with the same number of layers as GINAT, while adjusting the hidden dimension so that their parameter counts match as closely as possible.

**Graph-Describing Prompts Data.** We pair each graph dataset with a prompt dataset of approximately 200 prompts. As outlined in Section 1, these prompts describe the nature of the graph's data modality. This allows us to use the same prompt dataset for different datasets and tasks from the same data modality. For example, we use the same molecular graph-describing prompts for both OGBG − MOLHIV and ZINC. We split the prompt dataset randomly in an 8-1-1 ratio. In a training epoch, each graph sample is paired with a single prompt embedding drawn at random from the training split. For validation and test evaluation, prompts are drawn in fixed order from the corresponding splits. See Appendix H for listings of the used prompts.

**A Remark on Results.** Owing to resource constraints, we did not perform exhaustive hyperparameter tuning. Each dataset-method pair was tested under three layer depths and two widths, while several parameters (e.g., learning rate, scheduling, embedding dimensions, or different normalization layers) were left at default values. This may explain deviations between our baseline results and those published elsewhere. Nevertheless, our goal is to establish the relative advantage of GINAT.

Table 2: Social-network datasets accuracy (higher is better). GINAT results are highlighted in red. Best results on for each GNN-dataset pair are in **bold**. These graphs datasets are featureless; Baselines used random node features, while GINAT used constant node features.

| | REDDIT-BINARY | REDDIT-MULTI5K | REDDIT-MULTI12K | IMDB-BINARY | IMDB-MULTI |
|---|---|---|---|---|---|
| GCN | 0.785 ±0.06 | 0.512 ±0.04 | 0.456 ±0.01 | 0.607 ±0.04 | 0.371 ±0.01 |
| | **0.887** ±0.01 | **0.554** ±0.04 | **0.477** ±0.00 | **0.747** ±0.04 | **0.489** ±0.02 |
| GAT | 0.510 ±0.06 | 0.218 ±0.02 | 0.214 ±0.01 | 0.507 ±0.01 | 0.336 ±0.03 |
| | **0.930** ±0.01 | **0.552** ±0.02 | **0.493** ±0.01 | **0.753** ±0.05 | **0.478** ±0.02 |
| GATv2 | 0.510 ±0.06 | 0.198 ±0.02 | 0.214 ±0.01 | 0.500 ±0.00 | 0.327 ±0.03 |
| | **0.928** ±0.02 | **0.572** ±0.03 | **0.496** ±0.00 | **0.743** ±0.04 | **0.482** ±0.03 |
| GIN[E] | 0.935 ±0.02 | 0.550 ±0.03 | 0.498 ±0.01 | 0.723 ±0.02 | 0.422 ±0.04 |
| | **0.945** ±0.01 | **0.552** ±0.03 | **0.504** ±0.01 | **0.740** ±0.02 | **0.482** ±0.04 |
| GraphConv | 0.917 ±0.01 | 0.570 ±0.02 | 0.489 ±0.01 | 0.737 ±0.03 | 0.466 ±0.05 |
| | **0.927** ±0.03 | **0.572** ±0.03 | **0.498** ±0.01 | **0.753** ±0.05 | **0.491** ±0.03 |
| GraphGPS | 0.777 ±0.04 | 0.339 ±0.10 | 0.218 ±0.01 | 0.737 ±0.03 | 0.416 ±0.05 |
| | **0.928** ±0.01 | **0.510** ±0.01 | **0.477** ±0.03 | **0.750** ±0.04 | **0.442** ±0.04 |

## 5.2 GINAT IMPROVES GRAPH INFERENCE PERFORMANCE

In our main experiment, we seek to address (Q1) and establish the contribution of augmenting graph data with global, embedded textual descriptions of the graphs using GINAT.

**Settings.** We compare the performance of 7 well-known GNN baselines with their augmented counterparts on several graph-level datasets (an overview of the evaluated datasets is given in Appendix C). Specifically, we used standard MPNNs: GCN (Kipf & Welling, 2017), GAT (Veličković et al., 2018), GATv2 (Brody et al., 2022), GraphConv (Morris et al., 2019), GraphSage (Hamilton et al., 2017), and GIN (Xu et al., 2019) or GINE (Hu et al., 2020) when edge features were available. We also use the graph transformer GraphGPS (Rampášek et al., 2022) as a baseline. In these experiments, we used Qwen3-4B as the embedding LLM.

**Results.** The performance of GINAT against the baselines, shown in tables 1, 2 establish the effectiveness of GINAT across dataset sizes, tasks (binary and multiclass classification, regression), graph sizes and sparseness, and data domains. The configurations used for each dataset are listed in Appendix D.

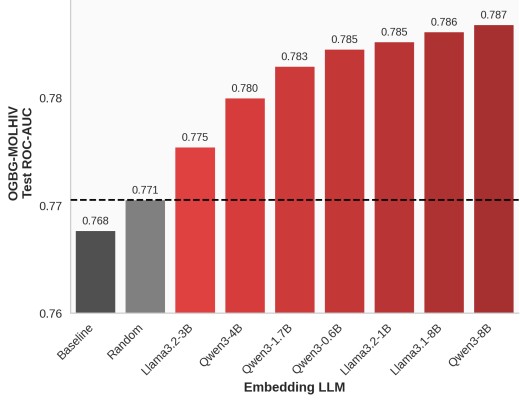

## 5.3 ABLATION STUDY: LLMs PROVIDE INFORMATIVE FEATURES

While our main experiment established that GINAT delivers performance gains, these improvements may not necessarily result from the inclusion of global features alone; factors such as injected randomness could also contribute to downstream performance. To investigate this, we tested GINAT with different LLMs embedding using the same prompts on an identical dataset and GNN configuration. Crucially, we also conducted a controlled experiment where

Figure 4: Performance of GINAT on MOLHIV with a GCN backbone, using various LLMs (all 5 layers, hidden size 512). All LLMs outperform the baseline and the random-noise control, marked by the black line. Larger models achieve the best results, supporting the claim that gains stem from meaningful prompt information.

the LLM embeddings were replaced by random Gaussian vectors of comparable size.

In this experiment, we train GINAT on the OGBG-MOLHIV dataset using GCN backbone. We evaluate GINAT using several LLMs: the 0.6B, 1.7B, 4B, and 8B versions of `Qwen3` (Yang et al., 2025), and `Llama3.1-8B`, `Llama3.2-3B`, and `Llama3.2-1B` (Grattafiori et al., 2024). All models were configured with 5 layers and a hidden dimension of 512. The results are illustrated in Figure 4, indicating that all tested LLMs yield substantial improvements over both the baseline and the random-noise controlled experiment. While the performance margin between models is not large, the strongest results are achieved with larger LLMs. This result provides further evidence that the gains stem from the information in the prompts - best captured by more expressive LLMs. Moreover, we see that LLMs consistently improve baseline performance.

## 5.4 NODE CLASSIFICATION: WHEN DO WE BENEFIT FROM GINAT?

Having established, through the first two experiments in Sections 5.2–5.3, both the effectiveness of LLM-embedded prompts and the capacity gains of GINAT, we next investigate its underlying properties. In particular, we seek to answer (Q3): is the contribution of GINAT *local* or *global*?

The answer is nontrivial: external features provide a global view of the graph, yet the embedded prompts interact locally with the GNN messages, rather than only after a $readout$ aggregation.

To shed light on this matter, we apply GINAT to two large-scale node-classification benchmarks: OGBN-ARXIV and OGBN-PRODUCTS, each accompanied by a suitable prompt dataset, as discussed in Appendix H. Below, we discuss the results, which are presented in Figure 5.

**Node-Level Settings.** We adopt experimental settings parallel to those of the graph-level settings. Specifically, for every batch of nodes sampled by GraphSAINT (Zeng et al., 2020), we draw a matching batch of prompts, and treat the sampled subgraph as a graph as in the graph-level settings.

**GINAT's Contribution is Global.** From Figure 5, it is evident that GINAT does not yield comparable gains on node-level tasks. This aligns with findings in the literature, which show that such tasks rely mainly on local neighborhood features rather than global information (Ying et al., 2019; Gasteiger et al., 2019; Wu et al., 2019; Hamilton et al., 2017). We therefore conclude that GINAT *supplements GNNs globally*. For additional insights and analysis of node-level results, we refer the reader to Appendix G.

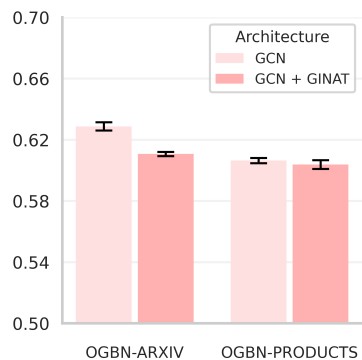

Figure 5: Performance of GINAT on node-classification benchmarks OGBN-ARXIV and OGBN-PRODUCTS. The results show that GINAT's contributes on a *global* level.

## 6 CONCLUSION

We introduced GINAT — a novel approach showing that graph representation learning can benefit from global graph descriptions, embedded and enriched by an LLM. GINAT leverages a cross-attention mechanism to integrate the two modalities and provably increases GNN capacity while maintaining permutation equivariance. Through extensive experiments across multiple domains and benchmarks, supported by newly constructed prompt datasets, we provided empirical evidence for the effectiveness of the approach. GINAT consistently improves performance on graph-level tasks, achieving gains around 10% improvement over strong baselines, while operating under the same parameter budget.

The theoretical and empirical gains in this work lay the foundation for numerous extensions and improvements, including alternative conjoining mechanisms and diverse prompting strategies.

**Limitations.** Currently, the improvements of our method are most pronounced in graph-level tasks, a consequence of the global nature of the prompts. We view extending the framework in this direction as an exciting opportunity for future research.

## AUTHORS' STATEMENTS

**Ethics Statements.** This work does not involve human subjects, personal or sensitive data, or applications in high-risk domains, to the best of our knowledge. All datasets used in our experiments are publicly available benchmark graph datasets (e.g., OGB) and were used in compliance with their respective licenses. The large language models employed in our study are publicly released open-source models obtained through HuggingFace. Our method is designed as a general framework for improving graph learning and does not target or promote harmful applications. We acknowledge that, as with any machine learning system, fairness and bias issues may arise if the approach is applied to sensitive data; however, such applications are beyond the scope of this paper.

**Reproducibility Statement.** We have taken care to ensure the reproducibility of our results. Complete details of datasets, model architectures, training settings, and hyperparameters are provided in the main text and Appendix. The prompt datasets used are documented in Appendix H. All baseline implementations follow the official PyTorch Geometric library (Fey & Lenssen, 2019), and pretrained LLMs are obtained from HuggingFace repositories. All experiments used $630, 1312, 15052010$ as seeds. Code for GINAT, including training scripts and evaluation pipelines, is available in the anonymised git repository `https://anonymous.4open.science/r/llm4gfm-3FBE/`

**Usage of Large Language Models in This Work.** LLMs were used in this work for several supportive purposes, including coding assistance, text editing, grammar refinement, and LaTeX formatting. In particular, they were employed to improve the clarity and coherence of the proofs accompanying our theoretical claims, and to expand the prompts datasets. Their use allowed the authors to focus more directly on research contributions rather than on ancillary editing tasks.

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

## A    EXPRESSIVENESS OF OUR APPROACH

In this appendix, we present a concrete example of two graphs indistinguishable by an MPNN that our approach successfully distinguishes. First, we prove that an MPNN, regardless of implementation, fails to distinguish these graphs. Then we show that our approach successfully distinguishes between them.

### A.1    MPNNs CANNOT DISTINGUISH BETWEEN TWO TRIANGLES AND AN HEXAGON

**Lemma 3.** *Let $G_1$ be the graph comprised of two disconnected triangles and $G_2$ the graph containing a 6-cycle. Let both graphs be initialized such that every node has the identical feature vector $x$. For any number of MPNN layers with any learned weights, and for any deterministic readout function $R$, the resulting graph embeddings will be identical: $H_{G_1} = H_{G_2}$*

*Proof.* We will show by induction that for any layer $l$, all 12 nodes across both graphs have the exact same embedding vector $h_v^{(k)}$.

It follows that the multiset of final node embeddings is identical as well,

$$\mathcal{H}_{G_1} = \left\{ \left\{ h_v^{(k)} \mid v \in G_1 \right\} \right\} = \left\{ \left\{ h_v^{(k)} \mid v \in G_2 \right\} \right\} = \mathcal{H}_{G_2}$$

Then, since the readout function $R$ is deterministic, the final embeddings hold

$$H_{G_1} = R(\mathcal{H}_{G_1}) = R(\mathcal{H}_{G_2}) = H_{G_2}$$

completing the proof.

**Base Case:** For $l = 0$, by definition, all nodes in both graphs are initialized with the same feature vector $x$.

**Inductive Step:** Assume that at layer $l - 1$, all nodes in both graphs have the same embedding, $\forall v \in G_1 \cup G_2, h_v^{(l-1)} = h^{(l-1)}$. The MPNN update is a function of the node's own embedding and the aggregation of its neighbors' embeddings:

$$h_v^{(l)} = update\left( h_v^{(l-1)}, aggregate\left( \left\{ \left( message\left( h_u^{(l-1)} \right) \right) \mid u \in N(v) \right\} \right) \right) \tag{8}$$

By definition, all nodes in $G_1, G_2$ have exactly two neighbors. Let $v$ be a node in one of the graphs. By the induction assumption, $v$ and its neighbors $u_1, u_2$, all have the same embedding $h^{(l-1)}$. The aggregated message to $v$ is then

$$m_v = aggregate\left( \{ message\left( h_{l-1} \right), message\left( h_{l-1} \right) \} \right) \tag{9}$$

Thus the new embedding $h_v^{(l)}$ is

$$h_v^{(l)} = update(h_{l-1}, m_v) \tag{10}$$

Since our choice of $v$ was arbitrary, this holds for all nodes in both graphs. $\square$

### A.2    PROOF OF LEMMA 1

For the reader's convenience, we reiterate the lemma here.

**Lemma 4** (GINAT Achieves Separation Given Informative Global Features)**.** *Let $G_1$ be the graph comprised of two disconnected triangles and $G_2$ the graph containing a 6-cycle. Let both graphs be initialized such that every node has an identical feature vector $x$. With each graph, we associate a global feature vector $g_1, g_2$, respectively. Redefine the message sent to each node during message passing as*

$$m_{vu}^{(l)} = message\left( h_u^{(l-1)} \right), \, u \in N(v)$$

$$m_{vu}'^{(l)} = m_{vu}^{(l)} + \text{CrossAttention}\left( m_{vu}^{(l)}, g \right) \tag{11}$$

*where $h_v^{(l)} = update\left(h_v^{(l-1)}, aggregate\left(\left\{\left(message\left(h_u^{(l-1)}\right)\right) \mid u \in N(v)\right\}\right)\right)$. For a 2-layer MPNN, there exists a choice of $g_1 \neq g_2$ such that the final readout of both graphs is distinct, $H_{G_1} \neq H_{G_2}$, and there exists a choice of $g_3 \neq g_4$ (which are not linearly dependent) such that the final readout of both graphs is identical, $H_{G_1} = H_{G_2}$.*

*Proof.* We prove each claim of the lemma constructively, and provide full calculations for the initial feature vector $x = [1, 3, 1, 2]^T$. For tractability, and without loss of generalization, we define $aggregate$ as summation and $update$ as a residual connection, i.e., $h_v^{(l)} = h_v^{(l-1)} + m_v'^{(l)}$. Similar proofs can be constructed for any MPNN-suitable selection of these functions.

**Proof of first claim** ($\exists g_1, g_2 : H_{G_1} \neq H_{G_2}$)   Let

$$g_1 = [3, 1, 0, 0]^T, g_2 = [6, 1, 0, 0]^T$$

The proof proceeds by tracing the computation.

All nodes in both graphs start with $h^{(0)} = x = [1, 3, 1, 2]^T$. Since all nodes have degree 2, the initial aggregated message is identical for all nodes:

$$m^{(1)} = x + x = 2x = [2, 6, 2, 4]^T \tag{12}$$

The attention mechanism then causes a divergence. For $G_1$, the dot product is

$$m^{(1)} \cdot g_1 = [2, 6, 2, 4] \cdot [3, 1, 0, 0]^T = 12 \tag{13}$$

The augmented message is

$$m_{G_1}' = m^{(1)} + 12g_1 = [2, 6, 2, 4]^T + [36, 12, 0, 0]^T = [38, 18, 2, 4]^T \tag{14}$$

For $G_2$, the dot product is

$$m^{(1)} \cdot g_2 = [2, 6, 2, 4] \cdot [6, 1, 0, 0]^T = 18 \tag{15}$$

The augmented message is

$$m_{G_2}' = m^{(1)} + 18g_2 = [2, 6, 2, 4]^T + [108, 18, 0, 0]^T = [110, 24, 2, 4]^T \tag{16}$$

Since the augmented messages are different, the updated node embeddings after the first layer are also different: $h_{G_1}^{(1)} = x + m_{G_1}' = [39, 21, 3, 6]^T$, while $h_{G_2}^{(1)} = x + m_{G_2}' = [111, 27, 3, 6]^T$.

As the node embeddings have diverged after the first layer, all subsequent computations will also differ, guaranteeing that the final multisets of node embeddings are distinct and thus the final readouts $H_{G_1} \neq H_{G_2}$.

**Proof of second claim** ($\exists g_3, g_4 : H_{G_1} = H_{G_2}$)   For brevity, we first compute the common messages that would be generated without attention:

$$m^{(1)} = 2x = [2, 6, 2, 4]^T \tag{17}$$

This gives

$$h^{(1)} = x + m^{(1)} = 3x = [3, 9, 3, 6]^T \tag{18}$$

The next message is

$$m^{(2)} = h^{(1)} + h^{(1)} = 2h^{(1)} = [6, 18, 6, 12]^T \tag{19}$$

Let

$$g_3 = [-3, 1, 0, 0]^T, g_4 = [-1, 0, 1, 0]^T$$

The proof proceeds by tracing. At layer 1, the initial message is $m^{(1)}$. The dot product for $G_1$ is

$$m^{(1)} \cdot g_1 = [2, 6, 2, 4] \cdot [-3, 1, 0, 0] = -6 + 6 = 0 \tag{20}$$

Similarly for $G_2$,

$$m^{(1)} \cdot g_2 = [2, 6, 2, 4] \cdot [-1, 0, 1, 0] = -2 + 2 = 0 \tag{21}$$

In both cases, the attention is nullified, so the augmented message is just $m^{(1)}$, and the updated embedding is $h^{(1)} = [3, 9, 3, 6]^T$ for all nodes in both graphs. At layer 2, the message is $m^{(2)}$. The

dot product for $G_1$ is

$$m^{(2)} \cdot g_3 = [6, 18, 6, 12] \cdot [-3, 1, 0, 0] = -18 + 18 = 0 \qquad (22)$$

For $G_2$,

$$m^{(2)} \cdot g_4 = [6, 18, 6, 12] \cdot [-1, 0, 1, 0] = -6 + 6 = 0 \qquad (23)$$

Again, the attention is nullified, and the augmented message is identical for both graphs. This ensures the final node embeddings are also identical. Since the final multisets of embeddings are identical, the readouts must be identical, $H_{G_1} = H_{G_2}$.

$\square$

## B  PRESERVATION OF EQUIVARIANCE

GNNs are inherently permutation equivariant: reordering the nodes in the input graph leads to the same reordering in the output node embeddings. Analogously, multi-modal cross-attention is also permutation equivariant with respect to the queries. In this section, we prove that our method maintains permutation equivariance - a key characteristic in any GNN.

**Proposition 5.** *Let $Q \in \mathbb{R}^{n_q \times d}$ denote the query matrix, $K \in \mathbb{R}^{n_k \times d}$ the key matrix, and $V \in \mathbb{R}^{n_k \times d_v}$ the value matrix. The cross-attention operator is defined as*

$$\mathrm{Attn}(Q, K, V) = \mathrm{softmax}\left(\frac{QK^\top}{\sqrt{d}}\right)V,$$

*where the softmax is applied row-wise. Then for any permutation matrix $P \in \{0, 1\}^{n_q \times n_q}$, it holds that*

$$\mathrm{Attn}(PQ, K, V) = P\,\mathrm{Attn}(Q, K, V).$$

*That is, cross-attention is permutation equivariant with respect to its queries.*

*Proof.* Consider the attention with permuted queries:

$$\mathrm{Attn}(PQ, K, V) = \mathrm{softmax}\left(\frac{(PQ)K^\top}{\sqrt{d}}\right)V.$$

By associativity of matrix multiplication,

$$(PQ)K^\top = P(QK^\top).$$

Since the softmax is applied row-wise, permuting the rows before softmax is equivalent to permuting them after softmax:

$$\mathrm{softmax}\big(P(QK^\top)\big) = P\,\mathrm{softmax}(QK^\top).$$

Therefore,

$$\mathrm{Attn}(PQ, K, V) = \big(P\,\mathrm{softmax}(QK^\top)\big)V = P\,\big(\mathrm{softmax}(QK^\top)V\big).$$

This simplifies to

$$\mathrm{Attn}(PQ, K, V) = P\,\mathrm{Attn}(Q, K, V),$$

proving permutation equivariance with respect to queries. $\square$

**Proposition 6.** *Standard GNNs are permutation equivariant: permuting the input node order results in the same permutation applied to the output embeddings. This property is preserved if the message function incorporates cross-attention.*

*Formally, consider a message passing layer of the form*

$$\tilde{h}_v^{(t+1)} = update\Big(h_v^{(t)},\ aggregate(\{m_{uv}^{(t)} : u \in \mathcal{N}(v)\}),\ \mathrm{CrossAttention}(h_v^{(t)}, H_T, H_T)\Big),$$

*where $h_v^{(t)}$ is the embedding of node $v$ at layer $t$, $\mathcal{N}(v)$ denotes its neighborhood, $m_{uv}^{(t)}$ is the message from node $u$ to $v$, AGG is a permutation-invariant aggregation function, CrossAttn is a cross-attention operator with $h_v^{(t)}$ as query and $H_T$ as keys/values, and $U$ is the node-wise update function. Then the resulting GNN layer is permutation equivariant with respect to the nodes of the input graph.*

*Proof.* Let $P$ be a permutation matrix applied to reorder the nodes in the input graph. Permutation equivariance requires that

$$f(PX, PAP^\top) = Pf(X, A),$$

where $X$ is the node feature matrix and $A$ the adjacency matrix.

Consider the message-passing layer. Each node embedding $h_v^{(t)}$ acts as a query to the cross-attention module, while $H_T$ provides the keys/values.

By Proposition 3, cross-attention is permutation equivariant with respect to its queries and permutation invariant with respect to the keys/values. Thus, applying $P$ permutes the queries consistently: $\text{CrossAttention}(Ph_v^{(t)}, H_T) = P\,\text{CrossAttention}(h_v^{(t)}, H_T)$. The aggregated messages from neighbors are permutation equivariant due to the permutation-invariance of $aggretate$, i.e., $aggregate(\{m_{p(u)p(v)}^{(t)} : u \in \mathcal{N}(v)\}) = P\,aggragate(\{m_{uv}^{(t)} : u \in \mathcal{N}(v)\})$. Finally, the node-wise update $update$ preserves the permutation, since it is applied independently to each node.

Combining these, each node update under permutation $P$ satisfies

$$\tilde{h}_{p(v)}^{(t+1)} = update\Big(Ph_v^{(t)}, P\,aggregate(\{m_{uv}^{(t)}\}), P\,\text{CrossAttention}(h_v^{(t)}, H_T)\Big) = P\tilde{h}_v^{(t+1)}.$$

Hence, the GNN layer with cross-attention in the message stage is permutation equivariant. $\qquad\square$

## C DATASETS INFORMATION

### C.1 MOLECULAR GRAPH DATASETS

Table 3: Summary statistics of molecular datasets. Feature columns report only dimensionality. Totals are estimated as #graphs $\times$ average edges.

| Dataset | Graphs | Nodes | Edges | Tot. E | Node dim | Edge dim |
|---|---|---|---|---|---|---|
| OGBG-MOLHIV | 41,127 | 25.5 | 27.5 | 1,130,992 | 9 | 3 |
| MUTAG | 188 | 17.9 | 19.8 | 3,721 | 1 | 1 |
| ENZYMES | 600 | 32.6 | 62.1 | 37,284 | 19 | – |
| PROTEINS | 1,113 | 39.1 | 72.8 | 81,049 | 2 | – |
| NCI1 | 4,110 | 29.9 | 32.3 | 132,753 | 1 | – |
| NCI109 | 4,127 | 29.7 | 32.1 | 132,601 | 1 | – |
| ZINC (full) | 249,456 | 23.2 | 24.9 | 6,211,454 | 1 | 1 |

**OGBG-MOLHIV.** A molecular property prediction benchmark: binary classification of whether a molecule inhibits HIV replication. Nodes are atoms with 9 categorical features (atomic number, chirality, aromaticity, etc.); edges are bonds with 3 categorical features (bond type, stereo, conjugation).

**MUTAG.** A dataset of nitroaromatic compounds labeled by mutagenicity. Nodes are atoms (atom type); edges are bonds (bond type).

**ENZYMES.** Contains 600 protein tertiary-structure graphs classified into 6 enzyme commission (EC) classes. Nodes are secondary structure elements (SSEs) with one categorical type (helix/sheet/turn) and 18 continuous attributes. Together, this yields 19 node dimensions. Edges connect interacting SSEs.

**PROTEINS.** Protein tertiary-structure graphs with binary labels (enzyme vs. non-enzyme). Nodes are SSEs with a categorical type and one continuous attribute (2 total dimensions). PROTEINS_full provides 29 attributes.

**NCI1 / NCI109.** Molecules tested for anti-cancer activity. Nodes are atoms (1 categorical type dimension); edges are unlabeled. Graph labels indicate activity against cancer cell lines.

**ZINC.** Large dataset of drug-like molecules from the ZINC database. Nodes are atoms (1 categorical type dimension), edges are bonds (1 categorical type dimension). Used primarily for property regression tasks.

### C.2 SOCIAL NETWORK GRAPH DATASETS

Table 4: Summary statistics of molecular datasets. Feature columns report only dimensionality. Totals are estimated as #graphs $\times$ average edges.

| Dataset | Graphs | Nodes | Edges | Tot. E | Node dim | Edge dim |
|---|---|---|---|---|---|---|
| OGBG-MOLHIV | 41,127 | 25.5 | 27.5 | 1,130,992 | 9 | 3 |
| MUTAG | 188 | 17.9 | 19.8 | 3,721 | 1 | 1 |
| ENZYMES | 600 | 32.6 | 62.1 | 37,284 | 19 | – |
| PROTEINS | 1,113 | 39.1 | 72.8 | 81,049 | 2 | – |
| NCI1 | 4,110 | 29.9 | 32.3 | 132,753 | 1 | – |
| NCI109 | 4,127 | 29.7 | 32.1 | 132,601 | 1 | – |
| ZINC (full) | 249,456 | 23.2 | 24.9 | 6,211,454 | 1 | 1 |

**REDDIT-BINARY, REDDIT-MULTI-5K, REDDIT-MULTI-12K.** Each graph represents a Reddit discussion thread. Nodes are users; edges connect users if one replied to the other. Graph labels correspond to subreddit/community categories (2, 5, and 11/12 classes, respectively).

**IMDB-BINARY / IMDB-MULTI.** Graphs are ego-networks of movie collaborations. Nodes are actors/actresses, and edges indicate co-appearance in a movie. Graph labels are genres: IMDB-BINARY distinguishes Action vs. Romance, while IMDB-MULTI has three genres (Comedy, Romance, Sci-Fi).

# D   DATASET TRAINING CONFIGURATION

As discussed in Section 5, for each dataset and GNN backbone pair, we searched for the optimal layer count and layer width combination. The table below details the configuration used for each dataset. We reiterate that the same configuration was used for the baseline model and the LLM augmented model.

Table 5: Configuration used for molecular graph datasets, given in a layers/hidden dimension format.

|             | GCN     | GraphConv | GraphSAGE | GIN     | GAT     | GATv2   | GraphGPS |
|-------------|---------|-----------|-----------|---------|---------|---------|----------|
| OGBG-MOLHIV | 5 / 256 | 6 / 256   | 6 / 512   | 4 / 512 | 5 / 512 | 5 / 512 | 4 / 512  |
| ZINC        | 5 / 512 | 5 / 512   | 4 / 512   | 5 / 256 | 4 / 256 | 4 / 256 | 5 / 256  |
| MUTAG       | 6 / 256 | 4 / 256   | 6 / 512   | 6 / 256 | 4 / 256 | 4 / 256 | 6 / 256  |
| PROTEINS    | 6 / 512 | 4 / 512   | 5 / 512   | 3 / 256 | 3 / 512 | 4 / 256 | 3 / 512  |
| ENZYMES     | 6 / 512 | 5 / 512   | 5 / 512   | 4 / 256 | 5 / 512 | 4 / 512 | 4 / 512  |
| NCI1        | 3 / 512 | 4 / 512   | 3 / 512   | 3 / 512 | 5 / 512 | 3 / 512 | 5 / 256  |
| NCI109      | 3 / 256 | 6 / 512   | 5 / 512   | 6 / 256 | 4 / 512 | 6 / 512 | 5 / 512  |

Table 6: Configuration used for social network graph datasets, given in a layers/hidden dimension format.

|                | GCN     | GraphConv | GIN     | GAT     | GATv2   | GraphGPS |
|----------------|---------|-----------|---------|---------|---------|----------|
| Reddit-Binary  | 4 / 256 | 5 / 512   | 5 / 512 | 5 / 512 | 5 / 256 | 4 / 512  |
| Reddit-Multi12k| 5 / 256 | 5 / 512   | 5 / 512 | 5 / 512 | 5 / 256 | 3 / 512  |
| Reddit-Multi5K | 4 / 256 | 4 / 256   | 3 / 512 | 5 / 256 | 5 / 256 | 5 / 256  |
| IMDB-Binary    | 4 / 512 | 4 / 512   | 6 / 256 | 5 / 256 | 5 / 512 | 4 / 256  |
| IMDB-Multi     | 6 / 256 | 5 / 256   | 4 / 256 | 5 / 256 | 6 / 512 | 4 / 512  |

# E CROSS-ATTENTION FOR GNN-LLM INTEGRATION

Cross-attention has emerged as a central mechanism for embedding and aligning multi-modal information, enabling models to selectively integrate representations across heterogeneous modalities. For example, they were proven effective in vision-language models (Li et al., 2023a; Alayrac et al., 2022), where textual tokens are grounded in visual features through learned cross-modal interactions.

Formally, given two node feature matrices $X_G \in \mathbb{R}^{n_g \times d_g}, X_T \in \mathbb{R}^{n_T \times d_t}$, the cross-attention layer is comprised of three (learned) weight matrices that project these features to a shared vector space with dimension $d$. Without loss of generality, we will use $X_G$ as queries and $X_T$ as keys and values denoted by $W_Q \in \mathbb{R}^{d_G \times d}, W_K \in \mathbb{R}^{d_T \times d}, W_V \in \mathbb{R}^{d_T, d}$. Then, we compute

$$Q = X_G W_Q^T \qquad\qquad K = X_T W_K^T \qquad\qquad V = X_T W_V^T$$

which are usually termed the query, key, and value matrices, respectively. The cross-attention operator is then defined as:

$$\text{CrossAttention}\,(X_G, X_T) = \text{softmax}\left(\frac{QK^\top}{\sqrt{d}}\right) V \tag{24}$$

In our GINAT, we explore cross-attention as an interface between GNNs and LLMs. Concretely, we consider GNNs that provide embeddings $H_G \in \mathbb{R}^{|V| \times d}$ that encode structural dependencies among nodes. Specifically, as described in Section 3, we use the messages passed between nodes as queries. As keys and values, we consider LLM-generated contextualized token embeddings $H_T \in \mathbb{R}^{L \times d}$. Cross-attention is then employed to align these spaces,

$$Z_G = \text{CrossAttention}(H_G, H_T), \tag{25}$$

which enriches the messages with semantic cues from language.

# F  BACKGROUND: MESSAGE PASSING NEURAL NETWORKS (MPNNS)

Let $G = (V, E)$ be a graph with node set $V$ and (directed or undirected) edge set $E \subseteq V \times V$. Each node $v \in V$ may have a feature vector $h_v^{(0)} \in \mathbb{R}^{d_0}$ and each edge $(u, v) \in E$ may have features $e_{uv} \in \mathbb{R}^{d_e}$. MPNNs produce layer-wise node representations $\{h_v^{(l)}\}_{v \in V}$ for $l = 0, \ldots, L$ by repeatedly exchanging *messages* along edges and updating local states. A single layer $l \to l+1$ is defined by three components:

$$message(\cdot), \quad aggregate(\cdot), \quad update(\cdot).$$

**Message.**   For each incoming edge $(u, v) \in E$, node $u$ sends a message to $v$:

$$m_{u \to v}^{(l)} = message\big(h_u^{(l)}, h_v^{(l)}, e_{uv}\big) \in \mathbb{R}^{d_m}. \tag{26}$$

The function *message* can be a neural network (e.g., MLP) or a parametric rule (e.g., linear map, attention score). It may depend on source features $h_u^{(l)}$, target features $h_v^{(l)}$, and edge features $e_{uv}$.

**Aggregate.**   Node $v$ aggregates all incoming messages from its neighborhood $\mathcal{N}(v) = \{u \mid (u, v) \in E\}$ using a permutation-invariant operator:

$$\bar{m}_v^{(l)} = aggregate\Big(\big\{\, m_{u \to v}^{(l)} : u \in \mathcal{N}(v) \,\big\}\Big). \tag{27}$$

Common choices include *sum*, *mean*, *max*, learned attention-weighted sums, or more expressive set functions. Permutation invariance ensures that reordering neighbors does not change the result.

**Update.**   Node $v$ updates its representation by combining its previous state with the aggregated message:

$$h_v^{(l+1)} = update\big(h_v^{(l)}, \bar{m}_v^{(l)}\big) \in \mathbb{R}^{d_{l+1}}. \tag{28}$$

Typical updates are MLPs, gated mechanisms (e.g., GRU), residual additions with normalization, or transformer-style blocks. With permutation-invariant *aggregate*, the mapping $H^{(l)} \mapsto H^{(l+1)}$ is *permutation equivariant* at the node level.

**Readout for graph-level tasks.**   For graph-level prediction, node embeddings after $L$ layers are pooled into a single graph representation via a permutation-invariant *readout*:

$$h_G = readout\Big(\big\{\, h_v^{(L)} : v \in V \,\big\}\Big), \qquad \hat{y}_G = f_{\text{head}}(h_G). \tag{29}$$

Common readouts include *sum/mean/max* pooling, attention pooling, Set2Set, SortPool, or hierarchical pooling. The invariance of *readout* guarantees that $\hat{y}_G$ is independent of node ordering.

**Design choices and examples.**

- **GCN-style layers.** Messages linearly transform neighbors, often with degree normalization, and *aggregate* is summation/mean; *update* applies a nonlinearity.
- **GAT-style layers.** *message* uses learned attention coefficients $\alpha_{uv}$ over neighbors; *aggregate* is the attention-weighted sum; *update* applies projection and nonlinearity.
- **Edge-aware MPNNs.** Edge features $e_{uv}$ modulate messages (e.g., via edge-conditioned filters).
- **Stabilization.** Residual/skip connections, normalization (batch/layer), and dropout are commonly used to improve optimization and mitigate oversmoothing.

**Properties.**   With permutation-invariant *aggregate* (and *readout*), MPNNs are permutation-equivariant at the node level and permutation-invariant at the graph level. Expressivity grows with depth and the choice of *message/aggregate/update*, and can be related to the Weisfeiler–Leman hierarchy under suitable conditions.

**Summary.**   An MPNN layer computes messages on edges (26), aggregates them per node (27), and updates node states (28); a graph-level predictor applies a permutation-invariant readout (29). This decomposition provides a flexible template that subsumes many popular GNN architectures while preserving the necessary symmetry constraints of graph data.

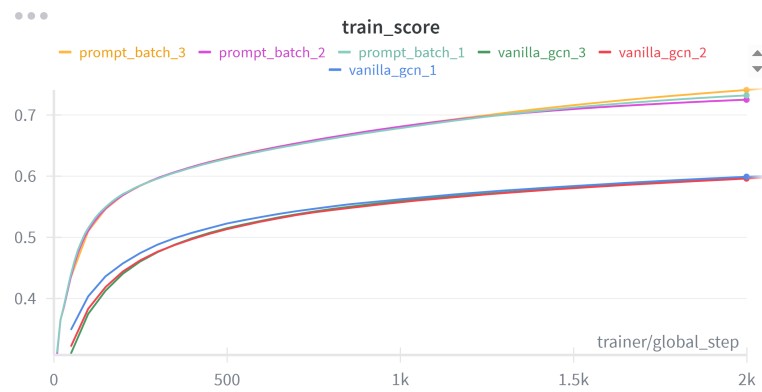

Figure 6: Batch of prompts to a single embedding for a batch of nodes configuration training fit curve for the OGBN-ARXIV dataset

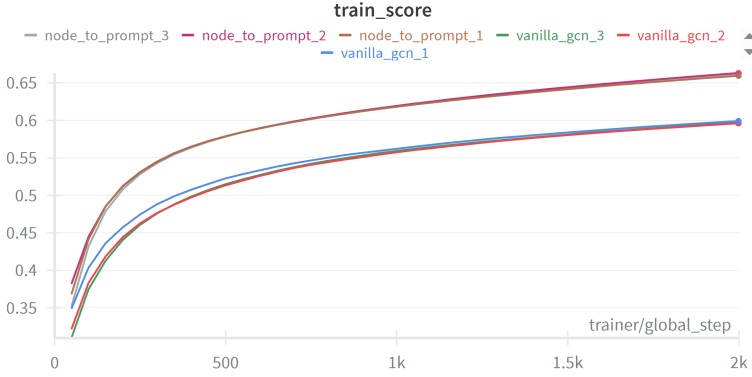

Figure 7: Prompt to node configuration training fit curve for the OGBN-ARXIV dataset

## G  NODE CLASSIFCATION

Here, we summarize the experiments conducted to validate the claim that global descriptive prompts do not significantly contribute to the model's generalization.

We designed two experimental setups to assess this claim. In the first approach, we employed the GraphSAINT Node Sampler to sample a batch of nodes from the graph. Analogous to the graph-level task, we also sampled a batch of prompts and extracted a single embedding representing their combined information. Cross-attention scores between these concise prompt embeddings and the node embeddings were then added as a bias to the node embeddings during the message-passing stage.

In the second approach, we adopted a prompt-per-node strategy, where each node is assigned a dedicated prompt. A cross-attention mechanism is then applied between the node and its corresponding prompt, allowing node-level interactions with the prompts.

We evaluated both approaches on the ogbn-arxiv and ogbn-products datasets and observed similar outcomes in both cases. Results, compared to a standard GCN, are shown in Figures 8. While the addition of prompts significantly improves the model's ability to fit the training data, Figure **??** demonstrates that they provide no meaningful improvement in generalization and, in some cases, even slightly degrade performance compared to a standard GCN.

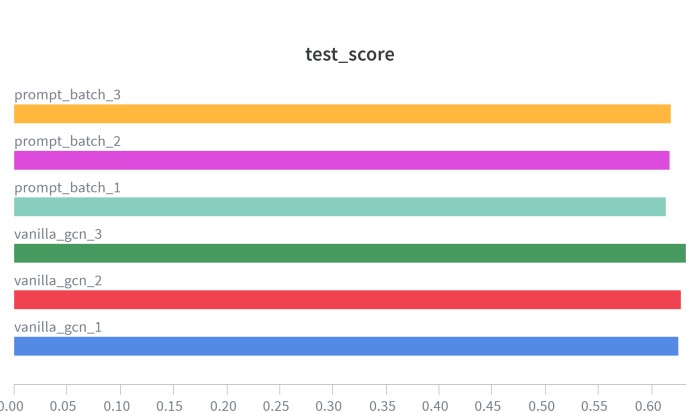

Figure 8: Batch of prompts to a single embedding for a batch of nodes configuration test results for the OGBN-ARXIV dataset

## H PROMPTS USED IN OUR EXPERIMENTS

In this appendix, we provide the prompts dataset we used to describe graphs in the reported experiments.

### H.1 MOLECULAR GRAPHS DESCRIPTIONS DATASET

```
[
"A molecular graph where each node represents an atom and each
edge a bond.",
"Graph where atoms are nodes and the bonds between them are
edges.",
"This graph models a molecule with atoms as nodes and bonds as
edges.",
"Represents a chemical structure where nodes are atoms and edges
are covalent bonds.",
"Each vertex is an atom, and each edge is a chemical bond.",
"A molecule's structure shown with atoms as nodes and bonds as
edges.",
"Nodes correspond to atoms, and edges represent the bonds
connecting them.",
"A graph illustrating atoms as vertices and their chemical
connections as edges.",
"Chemical bonds are edges connecting the atoms, which are
represented by nodes.",
"Atoms form the nodes of the graph, while bonds form the edges.",
"A representation of a molecule's atoms (nodes) and their bonds
(edges).",
"This graph depicts a molecule's atomic structure using nodes for
atoms, edges for bonds.",
"Vertices are atoms in a molecule, and edges are the bonds between
them.",
"A molecular graph where nodes stand for atoms and edges for
chemical bonds.",
"The graph's nodes are atoms; its edges are the bonds linking
them.",
"Modeling a chemical compound with atoms as nodes and bonds as
connecting edges.",
"A graph where nodes are atoms and edges show their covalent
connections.",
"Each node signifies an atom, and each edge signifies a chemical
bond.",
"This graph maps atoms to nodes and their chemical bonds to
edges.",
"A molecular representation using nodes for atoms and edges for
their bonds.",
"Atoms are modeled as nodes, and their covalent bonds are modeled
as edges.",
"A graph where vertices represent atoms and edges represent
chemical bonds.",
"This model uses nodes for atoms and edges for the bonds between
them.",
"Depicts a molecule where atoms are vertices and bonds are
edges.",
"A chemical graph with atoms as nodes and their connections as
edges.",
"Nodes are atoms, and the edges show the bonds that connect
them.",
```

```
"This graph's vertices are atoms, and its edges are chemical
bonds.",
"A representation where atoms are nodes and the edges are their
covalent bonds.",
"Molecular structure where atoms are nodes and edges are their
connecting bonds.",
"Graph showing atoms as nodes and the chemical bonds between them
as edges.",
"Atoms are represented by nodes, and bonds are represented by
edges.",
"A graph of a molecule where nodes are atoms and edges are
bonds.",
"The vertices of this graph are atoms; the edges are their
chemical bonds.",
"This graph illustrates a molecule's atoms as nodes and its bonds
as edges.",
"A model where nodes denote atoms and edges denote covalent
bonds.",
"Represents a molecule's atoms and bonds as a graph's nodes and
edges.",
"Each node is an atom, connected to others by edges representing
bonds.",
"A graph structure where nodes are atoms and edges are chemical
bonds.",
"The molecule's atoms are nodes, and the bonds between them are
edges.",
"A graph where each vertex is an atom and each edge a covalent
bond.",
"This graph's nodes correspond to atoms, and its edges to chemical
bonds.",
"A molecular structure is shown, with nodes as atoms and edges as
bonds.",
"Atoms are the vertices, and the bonds between them are the
edges.",
"A graph in which nodes are atoms and edges are their chemical
bonds.",
"This represents a molecule's structure with nodes for atoms,
edges for bonds.",
"Nodes stand for atoms, while edges stand for the bonds between
them.",
"A graph depicting atoms as nodes and their covalent bonds as
edges.",
"The atoms of a molecule are nodes; their bonds are the graph's
edges.",
"A model of a molecule using vertices for atoms and edges for
bonds.",
"This graph shows atoms as nodes and their bonds as connecting
edges.",
"A chemical structure where atoms are vertices and bonds are
edges.",
"Nodes are the atoms, and edges are the chemical bonds they
form.",
"Represents a molecule with atoms as vertices and bonds as
edges.",
"A graph where nodes are atoms and edges are the links between
them.",
"The vertices in this graph are atoms, and the edges are covalent
bonds.",
```

```
"A molecular graph where atoms are nodes and their bonds are the
edges.",
"This graph models atoms as vertices and their chemical bonds as
edges.",
"A representation where nodes are atoms and edges are their
chemical connections.",
"Atoms are depicted as nodes, and their bonds are depicted as
edges.",
"A graph illustrating a molecule with atoms as nodes and bonds as
edges.",
"Nodes represent the atoms, and edges represent the bonds in a
molecule.",
"This graph's nodes are a molecule's atoms, and its edges are the
bonds.",
"A model where vertices are atoms and edges are their covalent
bonds.",
"Depicts a chemical structure with nodes for atoms and edges for
bonds.",
"Atoms are the nodes, and the edges are the bonds that link
them.",
"A graph where each node is an atom and each edge is a bond.",
"The vertices represent atoms, and the edges represent the bonds
between them.",
"A molecular graph with nodes as atoms and edges as chemical
bonds.",
"This graph shows a molecule's atoms as nodes and their bonds as
edges.",
"Nodes are atoms in a chemical structure; edges are the bonds.",
"A graph where atoms are vertices and their covalent bonds are
edges.",
"This model shows atoms as nodes and the bonds connecting them as
edges.",
"Represents a molecule's atomic layout with nodes for atoms, edges
for bonds.",
"Each vertex corresponds to an atom, and each edge to a chemical
bond.",
"A graph of a molecule's atoms (vertices) and their bonds
(edges).",
"The nodes of this graph represent atoms, the edges represent
their bonds.",
"A molecular structure modeled with atoms as nodes and bonds as
edges.",
"This graph maps a molecule's atoms to nodes and its bonds to
edges.",
"A representation where vertices are atoms and edges are their
covalent bonds.",
"Atoms are shown as nodes, and their chemical bonds are shown as
edges.",
"A graph where the nodes are atoms and the edges are chemical
bonds.",
"This graph's vertices stand for atoms, and its edges stand for
bonds.",
"A model of a chemical structure with nodes as atoms, edges as
bonds.",
"Depicts atoms as nodes and the bonds that connect them as
edges.",
"A graph where each atom is a node and each bond is an edge.",
"The atoms are the graph's nodes, and their bonds are the edges.",
```

```
"This represents a molecule where atoms are nodes and bonds are
edges.",
"Nodes are atoms, and the edges between them represent covalent
bonds.",
"A graph showing a molecule's atoms as vertices and its bonds as
edges.",
"The vertices are atoms, and the edges are the chemical bonds
connecting them.",
"A molecular graph where nodes represent atoms and edges represent
their bonds.",
"This graph models a molecule's atoms as nodes and their bonds as
edges.",
"A representation of a chemical structure with atoms as nodes,
bonds as edges.",
"Atoms are mapped to nodes, and their bonds are mapped to edges.",
"A graph where nodes are atoms and edges are the connections
between them.",
"The vertices of the graph are atoms, and the edges are their
bonds.",
"Edges represent bonds between atoms, which are represented by
nodes.",
"Bonds are shown as edges, connecting the atoms which are the
nodes.",
"This graph's edges are bonds, and its nodes are the atoms they
connect.",
"Covalent bonds are edges, linking the atoms which are the
vertices.",
"Each edge is a chemical bond, and each vertex is an atom.",
"A graph where edges are bonds and nodes are the atoms they
link.",
"The bonds are edges, and the atoms they connect are the nodes.",
"Edges are covalent bonds connecting the atoms, which are the
nodes.",
"This graph models bonds as edges and the atoms they link as
nodes.",
"Represents chemical bonds as edges and atoms as the connected
nodes.",
"The edges are bonds, and the vertices are the atoms in the
molecule.",
"A model where edges are bonds and nodes are the atoms they
connect.",
"Bonds are the edges, connecting the atoms which are the graph's
nodes.",
"Edges are chemical bonds, and the nodes they connect are atoms.",
"This graph's edges represent bonds, and its nodes represent
atoms.",
"A representation where edges are bonds and nodes are the
connected atoms.",
"The chemical bonds are edges, and the atoms are the graph's
vertices.",
"Edges denote covalent bonds, and nodes denote the atoms they
link.",
"This graph shows bonds as edges and the atoms they connect as
nodes.",
"A structure where edges are bonds and the nodes are atoms.",
"Bonds are modeled as edges, and the atoms they connect are
nodes.",
"The edges of this graph are bonds; the nodes are the atoms.",
```

"A graph where bonds are edges and the atoms they connect are
nodes.",
"This model uses edges for bonds and nodes for the connected
atoms.",
"Depicts bonds as edges and the atoms they connect as vertices.",
"A chemical graph with bonds as edges and atoms as the nodes.",
"Edges are the bonds, and the nodes are the atoms they connect.",
"This graph's edges are chemical bonds, and its vertices are
atoms.",
"A representation where bonds are edges and the connected atoms
are nodes.",
"Molecular bonds are edges, and the atoms they connect are
nodes.",
"Graph showing bonds as edges and the atoms they connect as
nodes.",
"Bonds are represented by edges, and atoms are represented by
nodes.",
"A graph of a molecule where edges are bonds and nodes are
atoms.",
"The edges of this graph are chemical bonds; the vertices are
atoms.",
"This graph illustrates bonds as edges and its atoms as nodes.",
"A model where edges denote bonds and nodes denote atoms.",
"Represents a molecule's bonds and atoms as a graph's edges and
nodes.",
"Each edge is a bond, connecting atoms which are represented by
nodes.",
"A graph structure where edges are bonds and nodes are atoms.",
"The molecule's bonds are edges, and the atoms they connect are
nodes.",
"A graph where each edge is a covalent bond and each vertex is an
atom.",
"This graph's edges correspond to bonds, and its nodes to atoms.",
"A molecular structure is shown, with edges as bonds and nodes as
atoms.",
"Bonds are the edges, and the atoms they connect are the
vertices.",
"A graph in which edges are chemical bonds and nodes are atoms.",
"This represents a molecule's structure with edges for bonds,
nodes for atoms.",
"Edges stand for bonds, while nodes stand for the atoms they
connect.",
"A graph depicting bonds as edges and their connected atoms as
nodes.",
"The bonds of a molecule are edges; their atoms are the graph's
nodes.",
"A model of a molecule using edges for bonds and vertices for
atoms.",
"This graph shows bonds as edges and their connected atoms as
nodes.",
"A chemical structure where bonds are edges and atoms are
vertices.",
"Edges are the chemical bonds, and nodes are the atoms they
form.",
"Represents a molecule with bonds as edges and atoms as
vertices.",
"A graph where edges are the links and nodes are the atoms.",
"The edges in this graph are covalent bonds, and the vertices are
atoms.",

```
"A molecular graph where bonds are edges and their atoms are the
nodes.",
"This graph models bonds as edges and their chemical atoms as
vertices.",
"A representation where edges are chemical connections and nodes
are atoms.",
"Bonds are depicted as edges, and their atoms are depicted as
nodes.",
"A graph illustrating a molecule with bonds as edges and atoms as
nodes.",
"Edges represent the bonds, and nodes represent the atoms in a
molecule.",
"This graph's edges are a molecule's bonds, and its nodes are the
atoms.",
"A model where edges are covalent bonds and vertices are atoms.",
"Depicts a chemical structure with edges for bonds and nodes for
atoms.",
"Bonds are the edges, and the nodes are the atoms that they
link.",
"A graph where each edge is a bond and each node is an atom.",
"The edges represent the bonds, and the vertices represent the
atoms.",
"A molecular graph with edges as chemical bonds and nodes as
atoms.",
"This graph shows a molecule's bonds as edges and their atoms as
nodes.",
"Edges are the bonds in a chemical structure; nodes are the
atoms.",
"A graph where bonds are edges and their covalent atoms are
vertices.",
"This model shows bonds as edges and the atoms they connect as
nodes.",
"Represents a molecule's bond layout with edges for bonds, nodes
for atoms.",
"Each edge corresponds to a chemical bond, and each vertex to an
atom.",
"A graph of a molecule's bonds (edges) and their atoms
(vertices).",
"The edges of this graph represent bonds, the nodes represent
their atoms.",
"A molecular structure modeled with edges as bonds and nodes as
atoms.",
"This graph maps a molecule's bonds to edges and its atoms to
nodes.",
"A representation where edges are covalent bonds and vertices are
atoms.",
"Bonds are shown as edges, and their chemical atoms are shown as
nodes.",
"A graph where the edges are chemical bonds and the nodes are
atoms.",
"This graph's edges stand for bonds, and its vertices stand for
atoms.",
"A model of a chemical structure with edges as bonds, nodes as
atoms.",
"Depicts bonds as edges and the atoms that they connect as
nodes.",
"A graph where each bond is an edge and each atom is a node.",
"The bonds are the graph's edges, and their atoms are the nodes.",
```

```
"This represents a molecule where bonds are edges and atoms are
nodes.",
"Edges are covalent bonds, and the nodes between them represent
atoms.",
"A graph showing a molecule's bonds as edges and its atoms as
vertices.",
"The edges are the chemical bonds, and the vertices are the
connecting atoms.",
"A molecular graph where edges represent bonds and nodes represent
their atoms.",
"This graph models a molecule's bonds as edges and their atoms as
nodes.",
"A representation of a chemical structure with edges as bonds,
nodes as atoms.",
"Bonds are mapped to edges, and their atoms are mapped to nodes.",
"A graph where edges are the connections and nodes are the
atoms.",
"The edges of the graph are bonds, and the vertices are their
atoms."
]
```

## H.2  IMDB-* GRAPHS DESCRIPTIONS DATASET

```
[
  "Nodes are actors; an edge connects them if they appeared in the
  same movie together.",
  "An edge links two actor nodes if they both shared the screen in
  the same film.",
  "Actors who have co-starred in a movie are connected by an edge
  in this graph.",
  "This graph connects actors as nodes, an edge signifying a
  shared movie credit between them.",
  "When two actors appear in the same film, their corresponding
  nodes are linked by an edge.",
  "Vertices represent actors; a connection is formed if they
  collaborated on the same film project.",
  "Nodes are actors. An edge exists if two actors have worked on a
  movie together.",
  "An edge connects two nodes if the actors they represent
  appeared in the same motion picture.",
  "This network models actors as nodes, with edges for sharing a
  credit in a movie.",
  "If two actors were cast in the same movie, their nodes are
  connected by an edge.",
  "Actors are nodes. A link is drawn between them if they shared a
  film credit.",
  "Connect two actor nodes with an edge if they have ever
  performed in a movie together.",
  "This graph represents actors as nodes, linking those who have
  co-appeared in any film.",
  "A shared movie appearance creates an edge between the nodes
  representing the respective actors.",
  "Nodes are performers, and an edge links them if they acted in
  the same film production.",
  "An edge is created between two actor nodes if they were both in
  the same movie.",
  "The vertices are actors; an edge connects them if they were
  co-stars in a film.",
```

```
"Actors are represented as nodes, connected by edges if they
appeared in a shared movie.",
"Draw an edge between two actor nodes if they have a common film
in their filmography.",
"A connection exists between two actor nodes for a shared movie
appearance.",
"Nodes symbolize actors. An edge signifies that they have acted
together in a movie.",
"If actors share a movie credit, their nodes are connected in
this graph representation.",
"This is a graph of actors, where an edge connects any two who
acted together.",
"Two actor nodes are linked if the performers appeared together
in the same feature film.",
"An edge is present between two nodes if the actors shared a
role in a movie.",
"Actors who have collaborated on a film are connected by an edge
between their nodes.",
"Nodes represent actors. A link is formed if they have ever
co-starred in a film.",
"This network connects actors who have appeared in at least one
movie together.",
"An edge between two nodes means the actors performed in the
same film.",
"Let nodes be actors. An edge connects them if they shared a
movie production.",
"A graph where nodes are actors and an edge means a shared movie
credit.",
"Actors are nodes, linked by an edge if they appeared in the
same motion picture.",
"An edge joins two actor nodes if they have ever been in a movie
together.",
"Vertices are actors. A link between them means they acted in
the same movie.",
"Connect nodes if the actors they represent were featured in the
same movie.",
"Nodes are actors. They are connected if they have ever shared a
movie set.",
"An edge exists between actor nodes for every shared film
appearance.",
"The graph links actors as nodes if they have a common movie
credit.",
"Actors are the nodes, and a shared film role creates an edge
between them.",
"A shared movie credit between two actors results in an edge
connecting their nodes.",
"Nodes are actors. An edge connects two if they have a shared
movie in common.",
"If two actors were in the same movie, their nodes are connected
by a link.",
"This graph's nodes are actors. Edges connect those who have
acted in a movie together.",
"An edge connects two actor nodes if they have ever shared a
screen credit.",
"Actors are nodes. A connection is made if they have appeared in
a film together.",
"The nodes are actors, and an edge means they worked on the same
movie.",
```

```
"An edge links actor nodes that have co-appeared in any film
production.",
"Nodes represent actors. An edge is drawn for a shared
appearance in a movie.",
"Two nodes are connected if the actors they represent have
co-starred in a motion picture.",
"This network connects actors as nodes if they have ever worked
on a movie together.",
"A link is established between actor nodes if they were cast in
the same film.",
"The vertices are actors, connected by an edge if they were in
the same movie.",
"An edge exists between two nodes if their actors appeared in
the same film.",
"Nodes are actors. An edge means they have a common movie credit
on their resumes.",
"Connect two actor nodes if they have at any point acted in a
movie together.",
"A shared appearance in a film creates an edge between the
corresponding actor nodes.",
"This graph models actors as nodes, connecting them if they
co-starred in a movie.",
"An edge links actors who have worked together on any single
movie production.",
"Nodes represent actors. An edge signifies they were featured in
the same motion picture.",
"If actors have a shared movie credit, we connect their nodes
with an edge.",
"This is a network of actors, linked together if they appeared
in a movie.",
"Vertices are actors. An edge between them indicates a shared
film appearance.",
"Nodes are actors. A shared movie appearance between them
results in a connecting edge.",
"Connect two actor nodes if the performers have ever shared the
screen in a film.",
"An edge is created between actor nodes if they have a mutual
movie credit.",
"This graph links actors who have been cast in the same movie at
any time.",
"Nodes are actors. An edge connects them if they have a shared
film in their history.",
"A connection is made between two actor nodes if they acted in
the same movie.",
"If two actors worked together on a movie, their nodes are
linked in this graph.",
"An edge connects nodes of actors who have been co-stars in any
given movie.",
"The nodes are actors, and they are linked if they appeared in a
movie together.",
"Actors who have shared a film set are connected by an edge
between their nodes.",
"This network's nodes are actors. An edge means they appeared in
the same movie.",
"An edge links two actor nodes when they have a common movie
credit.",
"Nodes represent actors. An edge is formed for a shared role in
a film.",
```

```
"Connect nodes representing actors who have collaborated on a
motion picture.",
"An edge exists between two nodes if the actors have ever acted
together.",
"The graph has actors as nodes, with edges for shared movie
appearances.",
"A link between two nodes means the actors have a shared movie
credit.",
"Nodes are actors. An edge indicates they have been in a film
together.",
"Two actor nodes are connected if they have a mutual film in
their careers.",
"This graph connects actors as nodes if they have ever been in a
movie together.",
"An edge signifies that two actors have co-starred in the same
film production.",
"Vertices are actors. A connection exists if they were featured
in the same movie.",
"Nodes are actors. An edge is drawn if they have acted in a
movie together.",
"If two actors have a shared film credit, their nodes are
connected with an edge.",
"The network's nodes are actors, linked if they were cast in the
same movie.",
"An edge connects two actor nodes if they have ever worked on a
film together.",
"A shared movie role between two actors creates an edge
connecting their nodes.",
"This graph represents actors as nodes, linked if they have
shared a movie credit.",
"Nodes are actors. An edge is present if they have ever been in
a movie together.",
"Connect two nodes if the actors they represent have ever been
in a film together.",
"An edge links nodes of actors who have co-appeared in the same
movie.",
"A connection between actor nodes is formed if they have a
shared movie credit.",
"Nodes are actors. A link is created if they acted in the same
movie.",
"If two actors have worked on a film together, their nodes are
connected by an edge.",
"The vertices represent actors, connected by edges if they
shared a movie credit.",
"This graph connects actors as nodes who have ever been in a
movie together.",
"An edge between two nodes indicates a shared movie credit for
the actors.",
"Nodes are actors. They are connected if they have ever acted in
a film together.",
"An edge exists between two actor nodes if they have a shared
film in their filmography.",
"Connect nodes if their actors have appeared together in any
movie.",
"A link between two actor nodes means they have a mutual movie
credit.",
"This network has actors as nodes, with edges for co-starring in
a film.",
```

```
"Nodes are actors. An edge signifies that they have been in a
movie together.",
"Two nodes are connected if their respective actors have ever
been in a movie together.",
"An edge links actor nodes if they have ever collaborated on a
film project.",
"The graph's nodes are actors, connected if they have shared a
movie credit.",
"A shared appearance in a movie results in an edge between the
actor nodes.",
"Nodes are actors. An edge connects them if they have a mutual
movie appearance.",
"If two actors have been in a movie together, their nodes are
connected by an edge.",
"This graph links actors who have ever worked on the same movie
production.",
"An edge connects two actor nodes if they share a credit in any
film.",
"Vertices are actors. A link is made if they have been in a
movie together.",
"Nodes represent actors. An edge is created if they have a
shared movie credit.",
"Connect two nodes if the actors have ever been cast in the same
movie.",
"An edge means that two actors have at some point been in a
movie together.",
"The network connects actors as nodes if they have ever shared a
film credit.",
"A connection is established between actor nodes for a shared
movie role.",
"Nodes are actors. An edge is formed if they have ever been in a
film together.",
"If two actors have appeared in a movie together, their nodes
are linked.",
"This graph's vertices are actors, connected by edges for shared
movie roles.",
"An edge links two nodes if the actors have ever been in a movie
together.",
"Nodes are actors. A connection is made if they have a shared
movie credit.",
"Two nodes are linked if the actors have ever been in the same
movie.",
"An edge signifies that two actors have a shared credit in their
filmographies.",
"The graph connects actors who have at any time been in a movie
together.",
"A link between two nodes means the actors have been in a movie
together.",
"Nodes are actors. An edge is drawn between them if they have a
shared movie.",
"If actors have been in a movie together, their nodes are
connected by an edge.",
"This network represents actors as nodes, connected if they have
a shared film credit.",
"An edge between two nodes means the actors have a shared movie
credit.",
"Vertices are actors. A connection is made if they have acted in
a movie together.",
```

```
"Nodes are actors. An edge connects them if they have ever
worked together on a film.",
"Connect two actor nodes if they have ever been in the same film
production.",
"An edge links nodes if the actors have a shared movie
appearance.",
"The graph has actors as nodes. An edge means they were in a
movie together.",
"A shared film credit creates an edge between the corresponding
actor nodes.",
"Nodes are actors. A link is made if they have ever been in a
movie together.",
"If two actors have a shared film, an edge connects their
nodes.",
"This network connects actors as nodes if they have ever been in
a movie together.",
"An edge signifies that two actors have been in the same movie
together.",
"Vertices are actors. A connection is made if they have a shared
movie credit.",
"Nodes are actors. An edge is created if they have ever been in
a movie together.",
"Connect two nodes if their actors have a shared movie credit.",
"An edge means that two actors have been in a movie together.",
"The network connects actors who have ever been in a movie
together as nodes.",
"A connection is established between nodes if the actors were in
the same movie.",
"Nodes are actors. An edge is formed if they have been in a
movie together.",
"If two actors have been in a movie together, an edge connects
their nodes.",
"This graph's vertices are actors. Edges connect those with a
shared movie credit.",
"An edge links two nodes if their actors have ever been in a
movie together.",
"Nodes are actors. A connection is made if they share a movie
credit.",
"Two nodes are linked if their actors have ever been in the same
movie.",
"An edge signifies that two actors share a credit in their
filmographies.",
"The graph connects actors who have ever been in a movie
together.",
"A link between two nodes means the actors have been in a movie
together.",
"Nodes are actors. An edge is drawn if they have a shared
movie.",
"If actors have been in a movie together, their nodes are
connected.",
"This network represents actors as nodes, connected by a shared
film credit.",
"An edge between two nodes means the actors have a shared movie
credit.",
"Vertices are actors. A connection means they acted in a movie
together.",
"Nodes are actors. An edge connects them if they worked together
on a film.",
```

```
"Connect two actor nodes if they were in the same film
production.",
"An edge links nodes if the actors have a shared movie
appearance.",
"The graph has actors as nodes, with edges for being in a movie
together.",
"A shared film credit creates an edge between actor nodes.",
"Nodes are actors. A link is made if they were in a movie
together.",
"If two actors have a shared film, an edge connects their
nodes.",
"This network connects actors as nodes if they were in a movie
together.",
"An edge signifies that two actors were in the same movie
together.",
"Vertices are actors. A connection is made if they have a shared
movie credit.",
"Nodes are actors. An edge is created if they were in a movie
together.",
"Connect two nodes if their actors have a shared movie credit.",
"An edge means that two actors were in a movie together.",
"The network connects actors who were in a movie together as
nodes.",
"A connection is established between nodes if the actors were in
the same movie.",
"Nodes are actors. An edge is formed if they were in a movie
together.",
"If two actors were in a movie together, an edge connects their
nodes.",
"This graph's vertices are actors. Edges connect those with a
shared movie credit.",
"An edge links two nodes if their actors were in a movie
together.",
"Nodes are actors. A connection is made if they share a movie
credit.",
"Two nodes are linked if their actors were in the same movie.",
"An edge signifies that two actors share a credit in their
filmographies.",
"The graph connects actors who were in a movie together.",
"A link between two nodes means the actors were in a movie
together.",
"Nodes are actors. An edge is drawn if they have a shared
movie.",
"If actors were in a movie together, their nodes are
connected.",
"This network represents actors as nodes, connected by a shared
film credit.",
"An edge between two nodes means the actors have a shared movie
credit.",
"Vertices are actors. A connection means they acted in a movie
together.",
"Nodes are actors. An edge connects them if they worked together
on a film.",
"Connect two actor nodes if they were in the same film
production.",
"An edge links nodes if the actors have a shared movie
appearance.",
"The graph has actors as nodes, with edges for being in a movie
together.",
```

```
    "A shared film credit creates an edge between actor nodes.",
    "Nodes are actors. A link is made if they were in a movie
    together."
]
```

## H.3   REDDIT-* GRAPHS DESCRIPTIONS DATASET

```
[
  "This network maps comment-based relationships between users.
  Each node is a user, and a link signifies a direct reply.",
  "Construct a graph where participants are nodes. A tie exists if
  one member commented on another.",
  "An edge connects two user nodes when a comment is exchanged
  between the individuals they represent.",
  "We model a social graph where vertices are users. A directed
  arc from user A to B indicates A commented on B's content.",
  "Individuals are represented as nodes. A connection is formed
  between two nodes through a comment.",
  "This graph illustrates user engagement. Nodes represent users,
  and edges show who is commenting on whom.",
  "Each node corresponds to a platform user. Edges indicate a
  direct comment was made from one to another.",
  "A user is a node. We draw an edge between two nodes if a
  comment interaction occurs between them.",
  "This graph connects authors who comment on each other's work,
  where each author is a node in the network.",
  "Nodes are users. An edge is created if one user writes a remark
  to another.",
  "An edge is formed between two user nodes if a comment is
  exchanged. The nodes themselves simply represent the users.",
  "Let nodes be contributors. An edge connects nodes if a comment
  is made between them.",
  "Users are modeled as nodes. An edge between two nodes signifies
  a direct comment interaction.",
  "The network's vertices are users. An edge connects two vertices
  if one user commented on the other's content.",
  "Define nodes as users. An edge represents a comment from one
  user to another.",
  "A comment from one user to another creates an edge between
  their corresponding nodes in the graph.",
  "Nodes are individual users. An edge appears if one user has
  commented on the other.",
  "Represent users as nodes. An edge signifies that one of the
  users commented on the other.",
  "This is a graph of users where an edge connects two users if
  one has replied to the other.",
  "Vertices in the graph are users. A connection exists if one
  user commented directly on the other's post.",
  "When a user comments on another's content, an edge is created
  between their corresponding nodes.",
  "Nodes are users. An edge links two nodes if a comment occurred
  between those users.",
  "The graph maps user interactions. Nodes are users and edges are
  formed when one user comments on another.",
  "Each node is a user. An edge forms if one user comments on
  content created by the other.",
  "Model a network where users are nodes. A comment between two
  users creates an edge.",
```

```
"An edge is created between two user nodes upon a comment
interaction.",
"Let each user be a node. An edge exists if one user commented
on the other.",
"User nodes are connected by an edge if a comment was posted
from one to the other.",
"The graph represents users as nodes. An edge is present if a
comment was made from one user to another.",
"Nodes correspond to users, and edges represent comment
interactions between them.",
"A directed edge connects a commenter to a recipient, both of
whom are nodes representing users.",
"We construct a graph where nodes are users and an edge
indicates a comment was made.",
"Each node is a user. An edge connects two nodes if a comment
was exchanged between them.",
"Users are nodes in a graph. A link is formed if one user
replies to the other via comment.",
"This network connects users through comments. A node is a user
and an edge is a comment between them.",
"Nodes symbolize users. Edges are formed based on direct comment
interactions between these users.",
"An interaction via comment creates an edge between two user
nodes.",
"The vertices are users. An edge links two users if one has
commented on the other.",
"Users are represented as nodes in a graph, and an edge
signifies a comment interaction.",
"A graph is built where nodes are users. A comment from user A
to B creates a directed edge.",
"Nodes are users. An edge is created if one user directs a
comment at the other.",
"A comment between users forms an edge connecting their nodes.",
"Represent each user as a node. Connect two nodes with an edge
if one user commented on the other.",
"In this graph, a node is a user. An edge connects two users if
one has commented on the other.",
"We define a user interaction graph where nodes are users and
edges represent comments.",
"The graph's nodes are users. An edge is drawn between nodes if
their users had a comment interaction.",
"An edge signifies a comment from one user to another, where
each user is a node.",
"Let nodes be users. An edge from node A to B indicates user A
commented on user B.",
"Users are represented as nodes, and a comment between them
forms an edge.",
"The network's nodes are users. An edge exists if one user has
posted a comment to the other.",
"Nodes represent users. An edge connects two nodes if one user
has commented to the other.",
"A comment from one user to another establishes a connection
between their nodes.",
"In this representation, users are nodes. An edge is formed if
one user commented on another's content.",
"The graph links users who have commented on each other. The
nodes in the graph are the users themselves.",
"Each user is a node. A directed edge indicates a comment from
one user to another.",
```

"Nodes are users. An edge between two nodes means one commented on the other's post.",
"An edge connects two users if a comment was made between them. Each user is a node.",
"We create a graph where nodes are users, and edges are the comments made between them.",
"Users are nodes. An edge is present if one user comments on content posted by the other user.",
"This graph models comment interactions. Nodes are users, and an edge means a comment was made between them.",
"A node represents a user. An edge is created when a comment is directed from one user to another.",
"Nodes are users. An edge between two nodes indicates a comment was exchanged.",
"Let users be nodes. A directed edge connects a commenter to the recipient of the comment.",
"The graph's vertices are users, and an edge exists if a comment was made between them.",
"An edge is drawn between two user nodes if one commented on the other.",
"Nodes are users. An edge is added if one user comments on the other's activity.",
"Users are nodes in this network. An edge signifies a comment interaction.",
"We build a graph of users where an edge signifies a comment.",
"Represent users as nodes. If one comments on another, an edge connects their nodes.",
"This network connects users via comments, where nodes are users and edges are the comments.",
"Each node is a user, and a comment between them creates an edge.",
"Nodes are users, and edges denote comment interactions between them.",
"A user is a node. A directed edge is created from a user who comments to the one who receives it.",
"The graph structure has users as nodes, and a comment creates an edge between them.",
"If user A comments on user B, an edge links their respective nodes.",
"Nodes are users. An edge between them exists if one user commented on the other's post.",
"The graph's nodes represent users. An edge connects two nodes if a comment was made between them.",
"An edge indicates a comment from one user to another. The nodes in the graph are the users.",
"Users are nodes. An edge is established when one user comments on another's content.",
"This graph maps user interactions. Nodes are users and edges signify a comment between two users.",
"A node is a user. An edge is formed if a comment is made by one user to another.",
"Nodes represent users, and an edge is a comment between them.",
"Users are modeled as nodes. An edge signifies a comment directed from one user to another.",
"Vertices in the graph are users, and an edge connects them if a comment was exchanged.",
"An edge connects two user nodes following a comment interaction.",

```
"Nodes are users. A directed edge is drawn from the commenter to
the one being commented on.",
"We model a network where users are nodes, and comments are
edges.",
"A user is a node. An edge is drawn if one user has commented on
the other.",
"This graph connects users through their comments, where nodes
are users.",
"Nodes are users. An edge is created if one user directs a
comment at another.",
"An edge is formed between two nodes if their corresponding
users had a comment interaction.",
"Let nodes be users. An edge connects two nodes if one user
posted a comment to the other.",
"Users are the nodes of the graph, and a comment between them
creates an edge.",
"The network's nodes are users, and a comment forms an edge
between them.",
"A comment interaction between two users creates an edge between
their nodes.",
"Nodes are users. A link between two nodes means one user
commented on the other.",
"We construct a graph where users are nodes. A comment from one
to another creates an edge.",
"Each user is a node. An edge exists between nodes if their
users exchanged a comment.",
"Users are represented as nodes, and their comment interactions
are represented as edges.",
"The graph consists of user nodes. An edge is drawn if a comment
is made between them.",
"A directed edge from node A to B means user A commented on user
B.",
"Nodes represent users. An edge connects nodes if a comment was
made between the users.",
"An edge is created between two user nodes when one comments on
the other.",
"Let users be nodes. An edge is formed if one user comments on
another's content.",
"The graph models users as nodes and comments as edges.",
"A user is a node, and an edge is a comment between two such
nodes.",
"Nodes are users. An edge is drawn when a comment is made from
one user to another.",
"This network's vertices are users, connected by edges if a
comment was made.",
"An edge links two nodes if one user commented on the other. The
nodes represent users.",
"Users are nodes in a graph. An edge is created if one user
replies to another with a comment.",
"The graph maps user comments. Nodes are users and edges are
formed when one user comments on another.",
"A node is a user. An edge is established if a comment
interaction occurs between two users.",
"Nodes represent users, and edges signify comment
interactions.",
"Users are modeled as nodes. A directed edge connects a
commenter to the comment's recipient.",
"A graph is built with users as nodes and comments as edges.",
```

```
"A comment from user A to user B creates a link between their
nodes.",
"Nodes are users. An edge is present if one user has commented
on another's activity.",
"The graph's nodes are users. An edge is drawn if a comment
interaction happened between them.",
"An edge signifies a comment between two users, who are the
nodes of the graph.",
"Users are nodes. An edge appears when one user directs a
comment at another user.",
"This graph connects users who have engaged via comments. Nodes
are users and edges are comments.",
"A node is a user. An edge forms when one user comments on
another's content.",
"Nodes represent users, and an edge indicates a comment was made
between them.",
"Let users be nodes. A directed edge is created when one user
comments on another's post.",
"The vertices of the graph are users, and an edge signifies a
comment.",
"An edge connects two user nodes if a comment was exchanged
between them.",
"Nodes are users. An edge is added if one user comments on
another user's post.",
"This network shows user comment interactions. Nodes are users
and an edge means one commented on another.",
"We build a graph of users, where an edge represents a comment
between them.",
"Represent users as nodes. If one user comments on another,
their nodes are connected.",
"This network maps user comments. Nodes are users and an edge is
formed for each comment between users.",
"Each node is a user, and a comment creates an edge between
them.",
"Nodes are users, and edges represent comment-based
interactions.",
"A user is a node. A directed edge is created for a comment from
one user to another.",
"The graph structure uses users as nodes. A comment forms an
edge between them.",
"When user A comments on user B, an edge links their respective
nodes.",
"Nodes are users. An edge exists if one user commented on
another's contribution.",
"The graph's nodes are users. An edge is made between nodes if a
comment was exchanged.",
"An edge indicates a comment from one user to another. The nodes
are users.",
"Users are nodes. An edge is formed when one user comments on
something another user posted.",
"This graph connects users who comment on one another. Nodes are
users, and edges signify a comment interaction.",
"A node is a user. An edge is formed if a comment is directed
from one user to another.",
"Nodes represent users, and an edge is a comment.",
"Users are modeled as nodes. An edge is created if one user
comments on another's post.",
"Vertices in the graph are users, and an edge connects them if a
comment was made.",
```

"An edge connects two user nodes after a comment is made between
them.",
"Nodes are users. A directed edge goes from the commenter to the
person they commented on.",
"We model a network where users are nodes, and directed comments
are edges.",
"A user is a node. An edge is drawn if one user has commented on
another's content.",
"This graph links users via their comments. Nodes are users and
edges represent a comment between them.",
"Nodes are users. An edge is created if one user posts a comment
to another.",
"An edge is formed between two nodes if their users had a
comment exchange.",
"Let nodes be users. An edge connects two nodes if one user
wrote a comment to the other.",
"Users are the nodes of this graph, and a comment interaction
creates an edge.",
"The network's nodes are users, and a comment between them makes
an edge.",
"A comment interaction between users creates a link between
their nodes.",
"Nodes are users. A connection between two nodes means one user
commented to the other.",
"We construct a graph where users are nodes and a comment from
one to another creates a directed edge.",
"Each user is a node. An edge exists between nodes if a comment
was made between them.",
"Users are represented as nodes, and their direct comments as
edges.",
"The graph consists of user nodes. An edge is drawn if a comment
is posted between them.",
"A directed edge from node A to B indicates that user A
commented on user B, where A and B are users.",
"Nodes represent users. An edge connects nodes if a comment was
posted between the users.",
"An edge is created between two user nodes when one comments on
the other's post.",
"Let users be nodes. An edge is formed if one user comments on
another person's content.",
"The graph models users as nodes and directed comments as
edges.",
"A user is a node, and an edge is a comment interaction between
two user nodes.",
"Nodes are users. An edge is drawn when a comment is made from
one user to another.",
"This network's vertices are users, connected by edges if a
comment was made.",
"An edge links two nodes if one user commented on the other's
content, with nodes being users.",
"Users are nodes in a graph. An edge is created if one user
replies to another.",
"The graph maps user comment activity. Nodes are users and edges
are formed when one comments on another.",
"A node is a user. An edge is established if a comment
interaction occurs between two of them.",
"Nodes represent users, and edges signify direct comment
interactions between them.",

```
"Users are modeled as nodes. A directed edge connects a
commenter to the recipient of their comment.",
"A graph is built with users as nodes and their comments as the
edges between them.",
"A comment from user A to user B creates a link between their
corresponding nodes.",
"Nodes are users. An edge is present if one user has commented
on another user's activity.",
"The graph's nodes are users. An edge is drawn if a comment
interaction took place between them.",
"An edge signifies a comment between two distinct users, who are
represented as nodes.",
"Users are nodes. An edge appears when one user directs a
comment towards another user.",
"This graph connects users who have engaged with each other via
comments. Nodes are users, edges are comments.",
"A node is a user. An edge forms when one user comments on
another user's content.",
"Nodes represent users, and an edge indicates a comment was made
between them.",
"Let users be nodes. A directed edge is created when one user
comments on another user's post.",
"The vertices of the graph are users, and an edge signifies a
comment between them.",
"An edge connects two user nodes if a comment was exchanged
between those two users.",
"Nodes are users. An edge is added if one user comments on
another user's post.",
"This network displays user comment interactions, where nodes
are users.",
"We build a graph of users, where an edge represents a
comment.",
"Represent users as nodes. If one user comments on another,
their nodes are connected by an edge.",
"This network maps user comments. Nodes are users and an edge is
formed for each comment between them.",
"Each node is a user. A comment creates an edge between two of
them.",
"Nodes are users, and edges represent comment-based interactions
between them.",
"A user is a node. A directed edge is created for a comment from
one user to another.",
"The graph structure uses users as nodes. A comment forms an
edge between them.",
"When user A comments on user B, an edge links their respective
nodes.",
"Nodes are users. An edge exists if one user commented on
another's contribution to the platform.",
"The graph's nodes are users. An edge is made between nodes if a
comment was exchanged between them.",
"An edge indicates a comment from one user to another user. The
nodes are simply the users.",
"Users are nodes. An edge is formed when one user comments on
something another user has posted.",
"This graph connects users who comment on one another's posts,
where nodes are users.",
"A node is a user. An edge is formed if a comment is directed
from one user to another user.",
```

```
"Nodes represent users, and an edge represents a comment between
two users.",
"Users are modeled as nodes. An edge is created if one user
comments on another user's post.",
"Vertices in the graph are users, and an edge connects them if a
comment was made.",
"An edge connects two user nodes after a comment is made between
them.",
"Nodes are users. A directed edge goes from the commenter to the
person they commented on.",
"We model a network where users are nodes, and directed comments
form the edges.",
"A user is a node. An edge is drawn if one user has commented on
another."
]
```

## H.4   CITATION NETWORKS GRAPHS DESCRIPTIONS DATASET

```
[
    "The nodes correspond to academic papers. Edges reflect
    referenced literature connections.",
    "Nodes capture metadata about research manuscripts. Edges
    point to citation-based links.",
    "Each item represents a scholarly work. Edges tie together
    documents through citations.",
    "Edges indicate one paper citing another. These elements
    reflect authored academic contributions.",
    "These elements reflect authored academic contributions. Edges
    reflect referenced literature connections.",
    "Nodes denote individual research articles. Connections show
    references between documents.",
    "Each entry represents a scientific document. Edges reflect
    referenced literature connections.",
    "Nodes symbolize separate entries in a publication index.
    Links capture citation relationships.",
    "Nodes capture metadata about research manuscripts.
    Connections show references between documents.",
    "Each item represents a scholarly work. Connections show
    references between documents.",
    "The nodes mark distinct pieces of scientific writing. Links
    encode the act of citing prior work.",
    "Every node is tied to a research publication. Connections
    show references between documents.",
    "Nodes correspond to academic papers. Connections show
    references between documents.",
    "Nodes mark distinct pieces of scientific writing. Links
    encode the act of citing prior work.",
    "Each item represents a scholarly work. Connections show
    references between documents.",
    "Nodes denote individual research articles. Edges indicate one
    paper citing another.",
    "Every node is tied to a research publication. Links capture
    citation relationships.",
    "These nodes denote individual research articles. Links encode
    the act of citing prior work.",
    "Each item represents a scholarly work. Edges point to
    citation-based links.",
    "Nodes correspond to academic papers. Connections show
    references between documents.",
```

```
"Nodes symbolize separate entries in a publication index.
Edges reflect referenced literature connections.",
"Nodes denote individual research articles. Links capture
citation relationships.",
"Each node holds information about a published study. Links
encode the act of citing prior work.",
"Nodes mark distinct pieces of scientific writing. Edges tie
together documents through citations.",
"Each entry represents a scientific document. Links capture
citation relationships.",
"Every node is tied to a research publication. Edges tie
together documents through citations.",
"Each item represents a scholarly work. Links encode the act
of citing prior work.",
"Nodes denote individual research articles. Links encode the
act of citing prior work.",
"Each entry represents a scientific document. Edges denote
academic referencing paths.",
"Each node holds information about a published study. Edges
point to citation-based links.",
"These elements reflect authored academic contributions. Links
capture citation relationships.",
"Each entry represents a scientific document. Edges reflect
referenced literature connections.",
"Nodes mark distinct pieces of scientific writing. Edges
reflect referenced literature connections.",
"Every node is tied to a research publication. Links encode
the act of citing prior work.",
"Nodes capture metadata about research manuscripts.
Connections show references between documents.",
"These elements reflect authored academic contributions.
Connections arise from inter-paper citations.",
"Nodes correspond to academic papers. Edges reflect referenced
literature connections.",
"Each node holds information about a published study. Edges
indicate one paper citing another.",
"Every node is tied to a research publication. Edges denote
academic referencing paths.",
"Nodes denote individual research articles. Connections arise
from inter-paper citations.",
"Each node holds information about a published study. Links
capture citation relationships.",
"Nodes symbolize separate entries in a publication index.
Links capture citation relationships.",
"Nodes correspond to academic papers. Edges denote academic
referencing paths.",
"Nodes denote individual research articles. Links encode the
act of citing prior work.",
"Each item represents a scholarly work. Connections arise from
inter-paper citations.",
"Every node is tied to a research publication. Links capture
citation relationships.",
"Nodes symbolize separate entries in a publication index.
Connections show references between documents.",
"Nodes denote individual research articles. Edges point to
citation-based links.",
"Each item represents a scholarly work. Connections show
references between documents.",
```

```
"Nodes mark distinct pieces of scientific writing. Edges
reflect referenced literature connections.",
"Nodes correspond to academic papers. Links encode the act of
citing prior work.",
"Each entry represents a scientific document. Connections show
references between documents.",
"Each node holds information about a published study. Links
encode the act of citing prior work.",
"Nodes denote individual research articles. Links encode the
act of citing prior work.",
"Nodes symbolize separate entries in a publication index.
Edges denote academic referencing paths.",
"Nodes capture metadata about research manuscripts. Links
encode the act of citing prior work.",
"Each item represents a scholarly work. Edges tie together
documents through citations.",
"Each node holds information about a published study. Edges
denote academic referencing paths.",
"Nodes correspond to academic papers. Edges point to
citation-based links.",
"These elements reflect authored academic contributions. Edges
point to citation-based links.",
"Each node holds information about a published study. Edges
reflect referenced literature connections.",
"Each entry represents a scientific document. Edges indicate
one paper citing another.",
"Nodes mark distinct pieces of scientific writing. Connections
show references between documents.",
"Nodes denote individual research articles. Links encode the
act of citing prior work.",
"Each node holds information about a published study. Links
capture citation relationships.",
"Nodes correspond to academic papers. Connections show
references between documents.",
"Each entry represents a scientific document. Links capture
citation relationships.",
"Each node holds information about a published study. Edges
denote academic referencing paths.",
"Nodes denote individual research articles. Edges reflect
referenced literature connections.",
"Each item represents a scholarly work. Edges reflect
referenced literature connections.",
"Each node holds information about a published study.
Connections show references between documents.",
"Each entry represents a scientific document. Edges denote
academic referencing paths.",
"Nodes symbolize separate entries in a publication index.
Connections arise from inter-paper citations.",
"Nodes correspond to academic papers. Links encode the act of
citing prior work.",
"Nodes capture metadata about research manuscripts.
Connections arise from inter-paper citations.",
"Each item represents a scholarly work. Edges indicate one
paper citing another.",
"Each node holds information about a published study. Edges
indicate one paper citing another.",
"Nodes correspond to academic papers. Links capture citation
relationships.",
```

```
"Each entry represents a scientific document. Edges denote
academic referencing paths.",
"Nodes denote individual research articles. Connections arise
from inter-paper citations.",
"Nodes capture metadata about research manuscripts. Links
encode the act of citing prior work.",
"Nodes symbolize separate entries in a publication index.
Edges tie together documents through citations.",
"Each item represents a scholarly work. Edges point to
citation-based links.",
"Every node is tied to a research publication. Edges denote
academic referencing paths.",
"Each node holds information about a published study. Edges
point to citation-based links.",
"Nodes denote individual research articles. Links capture
citation relationships.",
"Each item represents a scholarly work. Connections arise from
inter-paper citations.",
"Nodes symbolize separate entries in a publication index.
Edges indicate one paper citing another.",
"Each entry represents a scientific document. Edges reflect
referenced literature connections.",
"Nodes correspond to academic papers. Edges denote academic
referencing paths.",
"Nodes capture metadata about research manuscripts.
Connections arise from inter-paper citations.",
"Each item represents a scholarly work. Links capture citation
relationships.",
"Every node is tied to a research publication. Connections
show references between documents.",
"Nodes denote individual research articles. Connections show
references between documents.",
"Nodes symbolize separate entries in a publication index.
Connections arise from inter-paper citations.",
"Nodes capture metadata about research manuscripts. Edges tie
together documents through citations.",
"Each node holds information about a published study. Edges
tie together documents through citations.",
"Each entry represents a scientific document. Connections
arise from inter-paper citations.",
"Each node holds information about a published study.
Connections arise from inter-paper citations.",
"Nodes symbolize separate entries in a publication index.
Edges reflect referenced literature connections.",
"Nodes correspond to academic papers. Edges reflect referenced
literature connections.",
"Each item represents a scholarly work. Links encode the act
of citing prior work.",
"Nodes denote individual research articles. Connections arise
from inter-paper citations.",
"Each node holds information about a published study.
Connections show references between documents.",
"Edges tie together documents through citations. These nodes
denote individual research articles.",
"Each connection maps a citation trace. Every node holds
information about a published study.",
"Edges reflect referenced literature connections. Every node
holds information about a published study.",
```

```
"Connections arise from inter-paper citations. These nodes
denote individual research articles.",
"Connections arise from inter-paper citations. Every node is
tied to a research publication.",
"Connections show references between documents. These elements
reflect authored academic contributions.",
"Edges tie together documents through citations. The nodes
mark distinct pieces of scientific writing.",
"Links capture citation relationships. Nodes capture metadata
about research manuscripts.",
"Connections show references between documents. Each entry
represents a scientific document.",
"Links capture citation relationships. Each entry represents a
scientific document.",
"Connections arise from inter-paper citations. These nodes
denote individual research articles.",
"Edges reflect referenced literature connections. Every node
is tied to a research publication.",
"Edges point to citation-based links. The nodes mark distinct
pieces of scientific writing.",
"Edges denote academic referencing paths. Each item represents
a scholarly work.",
"Links capture citation relationships. These elements reflect
authored academic contributions.",
"Edges point to citation-based links. Nodes symbolize separate
entries in a publication index.",
"Connections show references between documents. These nodes
denote individual research articles.",
"Links capture citation relationships. Nodes symbolize
separate entries in a publication index.",
"Links encode the act of citing prior work. Nodes symbolize
separate entries in a publication index.",
"Edges indicate one paper citing another. Nodes symbolize
separate entries in a publication index.",
"Links capture citation relationships. Nodes capture metadata
about research manuscripts.",
"Edges reflect referenced literature connections. Nodes
symbolize separate entries in a publication index.",
"Links encode the act of citing prior work. Each item
represents a scholarly work.",
"Links capture citation relationships. Nodes symbolize
separate entries in a publication index.",
"Links capture citation relationships. Every node holds
information about a published study.",
"Links capture citation relationships. Every node is tied to a
research publication.",
"Connections arise from inter-paper citations. These nodes
denote individual research articles.",
"Edges indicate one paper citing another. Nodes capture
metadata about research manuscripts.",
"Links encode the act of citing prior work. The nodes
correspond to academic papers.",
"Links encode the act of citing prior work. These elements
reflect authored academic contributions.",
"Edges indicate one paper citing another. Every node is tied
to a research publication.",
"Links capture citation relationships. The nodes correspond to
academic papers.",
```

```
"Each connection maps a citation trace. Every node is tied to
a research publication.",
"Edges tie together documents through citations. Every node
holds information about a published study.",
"Connections show references between documents. The nodes mark
distinct pieces of scientific writing.",
"Edges denote academic referencing paths. Each item represents
a scholarly work.",
"Links encode the act of citing prior work. Each entry
represents a scientific document.",
"Edges denote academic referencing paths. These nodes denote
individual research articles.",
"Links capture citation relationships. The nodes correspond to
academic papers.",
"Links encode the act of citing prior work. The nodes
correspond to academic papers.",
"Each connection maps a citation trace. These nodes denote
individual research articles.",
"Links encode the act of citing prior work. The nodes
correspond to academic papers.",
"Links capture citation relationships. Each entry represents a
scientific document.",
"Edges point to citation-based links. Each entry represents a
scientific document.",
"Edges point to citation-based links. Nodes capture metadata
about research manuscripts.",
"Each connection maps a citation trace. Every node holds
information about a published study.",
"Connections show references between documents. Every node is
tied to a research publication.",
"Edges tie together documents through citations. Each item
represents a scholarly work.",
"Connections arise from inter-paper citations. These elements
reflect authored academic contributions.",
"Each connection maps a citation trace. These elements reflect
authored academic contributions.",
"The graph represents academic documents linked through
references. Every node is tied to a research publication.
Edges point to citation-based links.",
"The structure is a scholarly graph of interconnected
research. Each item represents a scholarly work. Edges point
to citation-based links.",
"The graph models a community of papers and their references.
Each item represents a scholarly work. Each connection maps a
citation trace.",
"The structure is a scholarly graph of interconnected
research. These elements reflect authored academic
contributions. Links encode the act of citing prior work.",
"This is a directed graph capturing citation flows among
articles. Each entry represents a scientific document. Edges
tie together documents through citations.",
"This is a network of publications with citation
relationships. The nodes correspond to academic papers.
Connections show references between documents.",
"A citation web connects scholarly articles in this dataset.
These nodes denote individual research articles. Edges tie
together documents through citations.",
```

```
"This dataset builds a graph of paper citations. Every node
holds information about a published study. Each connection
maps a citation trace.",
"The graph models a community of papers and their references.
Every node is tied to a research publication. Links capture
citation relationships.",
"A citation web connects scholarly articles in this dataset.
Every node holds information about a published study. Edges
indicate one paper citing another.",
"The graph represents academic documents linked through
references. These nodes denote individual research articles.
Connections arise from inter-paper citations.",
"This dataset forms a citation network of scientific
publications. Every node is tied to a research publication.
Each connection maps a citation trace.",
"The graph models a community of papers and their references.
Nodes symbolize separate entries in a publication index. Edges
reflect referenced literature connections.",
"The structure is a scholarly graph of interconnected
research. Each entry represents a scientific document.
Connections arise from inter-paper citations.",
"A network where nodes are papers and edges are citations.
These nodes denote individual research articles. Connections
show references between documents.",
"A citation web connects scholarly articles in this dataset.
Each item represents a scholarly work. Connections arise from
inter-paper citations.",
"This dataset forms a citation network of scientific
publications. Every node is tied to a research publication.
Edges denote academic referencing paths.",
"A network where nodes are papers and edges are citations.
These elements reflect authored academic contributions. Edges
denote academic referencing paths.",
"This structure captures how academic works cite each other.
Each entry represents a scientific document. Connections show
references between documents.",
"A network where nodes are papers and edges are citations.
Every node holds information about a published study. Edges
reflect referenced literature connections.",
"This is a network of publications with citation
relationships. These elements reflect authored academic
contributions. Edges indicate one paper citing another.",
"The structure is a scholarly graph of interconnected
research. Every node holds information about a published
study. Edges tie together documents through citations.",
"The graph represents academic documents linked through
references. The nodes correspond to academic papers.
Connections show references between documents.",
"A citation web connects scholarly articles in this dataset.
Every node is tied to a research publication. Edges reflect
referenced literature connections.",
"This dataset builds a graph of paper citations. Nodes
symbolize separate entries in a publication index. Edges
reflect referenced literature connections.",
"This dataset builds a graph of paper citations. The nodes
correspond to academic papers. Connections arise from
inter-paper citations.",
```

```
"The structure is a scholarly graph of interconnected
research. Each item represents a scholarly work. Edges denote
academic referencing paths.",
"A citation web connects scholarly articles in this dataset.
The nodes correspond to academic papers. Edges indicate one
paper citing another.",
"This is a directed graph capturing citation flows among
articles. The nodes correspond to academic papers. Edges point
to citation-based links.",
"This is a network of publications with citation
relationships. Each entry represents a scientific document.
Edges reflect referenced literature connections.",
"The graph models a community of papers and their references.
These elements reflect authored academic contributions. Edges
denote academic referencing paths.",
"The graph models a community of papers and their references.
Nodes symbolize separate entries in a publication index. Edges
tie together documents through citations.",
"A citation web connects scholarly articles in this dataset.
Each item represents a scholarly work. Edges point to
citation-based links.",
"The graph represents academic documents linked through
references. Nodes capture metadata about research manuscripts.
Each connection maps a citation trace.",
"The graph represents academic documents linked through
references. These nodes denote individual research articles.
Each connection maps a citation trace.",
"The graph models a community of papers and their references.
Nodes symbolize separate entries in a publication index.
Connections arise from inter-paper citations.",
"The graph models a community of papers and their references.
These elements reflect authored academic contributions. Links
encode the act of citing prior work.",
"The structure is a scholarly graph of interconnected
research. Nodes symbolize separate entries in a publication
index. Edges reflect referenced literature connections.",
"This dataset builds a graph of paper citations. The nodes
correspond to academic papers. Edges denote academic
referencing paths.",
"This structure captures how academic works cite each other.
The nodes correspond to academic papers. Edges reflect
referenced literature connections.",
"This is a directed graph capturing citation flows among
articles. Every node is tied to a research publication.
Connections show references between documents.",
"A citation web connects scholarly articles in this dataset.
Every node is tied to a research publication. Connections
arise from inter-paper citations.",
"This structure captures how academic works cite each other.
These nodes denote individual research articles. Edges
indicate one paper citing another.",
"A network where nodes are papers and edges are citations.
Each item represents a scholarly work. Edges point to
citation-based links.",
"A network where nodes are papers and edges are citations.
Every node holds information about a published study.
Connections arise from inter-paper citations.",
```

```
      "This is a directed graph capturing citation flows among
      articles. Each item represents a scholarly work. Edges
      indicate one paper citing another."
]
```

## H.5 OGBN-Products Graphs Descriptions Dataset

```
[
      "the network's nodes are products, and the edges connecting
      them show that those products are often bought in the same
      purchase",
      "in this graph, products are represented by nodes, and a
      shared purchase is shown by an edge between two products",
      "edges link products (nodes) that are purchased
      simultaneously, while nodes signify the products themselves",
      "a node stands for a product, and an edge between two nodes
      means those two products are frequently part of the same
      transaction",
      "products are the nodes in this network, and an edge between
      any two of them signifies they are commonly bought together",
      "this model uses nodes for products and edges to represent
      joint purchases of those products",
      "each product is a node, and an edge is drawn between two
      products when they are purchased in a single transaction",
      "nodes are individual products, and an edge connecting any two
      nodes shows they are bought as a pair",
      "if products are nodes, then a connecting edge indicates that
      those products were purchased at the same time",
      "nodes in this graph are products, and their shared purchase
      is symbolized by an edge linking them",
      "products sold are represented by nodes, and edges show which
      pairs of products were acquired in the same purchase",
      "in this system, products are nodes, and an edge exists if two
      products are bought together",
      "the nodes are products, and an edge between them denotes a
      co-purchase of those items",
      "a product is a node, and an edge between two nodes indicates
      a simultaneous purchase of the products",
      "nodes symbolize products, and an edge between them represents
      that they are typically purchased as a set",
      "products are nodes, and an edge connecting them signifies
      that they are frequently purchased together in one order",
      "this network uses nodes for products and edges to indicate
      that two products are found in the same transaction",
      "each product is a node, and an edge between two nodes
      signifies that the products were purchased together",
      "the nodes in this graph are products, and the edges represent
      a relationship of being bought at the same time",
      "products are nodes, and an edge between them means they are
      part of a single purchase",
      "nodes stand for products, while edges link any two products
      that were bought in a single transaction",
      "products are represented by nodes, and edges connecting those
      nodes signify they are often purchased together",
      "the nodes are products, and the edges show a connection based
      on them being bought in the same transaction",
      "a node represents a product, and an edge between two products
      shows they are purchased together",
```

```
        "nodes symbolize products, and edges connect those products
        that are found together in a single sale",
        "in this model, products are nodes, and edges are drawn to
        indicate which products are purchased together",
        "nodes are products sold, and an edge between two nodes
        indicates they are included in the same purchase",
        "the network's nodes are products, and edges connect products
        that are commonly purchased in the same transaction",
        "an edge between two product nodes means those items were
        purchased at the same time",
        "nodes are products, and an edge is drawn between products
        that are frequently purchased together",
        "a node represents a product, and an edge between two nodes
        shows they are frequently purchased together",
        "the nodes of this graph are products, and the edges signify
        that two products were bought as a pair",
        "products are nodes, and an edge indicates a joint purchase of
        those products",
        "a product is a node, and an edge between two nodes means they
        are co-purchased",
        "the nodes represent products, and the edges connect products
        that are bought together",
        "nodes are products, and a link between two products means
        they were purchased as a set",
        "in this representation, products are nodes, and edges between
        them represent simultaneous purchases",
        "products are the nodes, and a connecting edge signifies that
        they are bought in the same transaction",
        "nodes symbolize products, and an edge is created when those
        products are bought together",
        "the nodes are products sold, and the edges indicate which
        products are purchased together",
        "in this network, products are nodes, and an edge shows that
        they are co-purchased",
        "nodes stand for products, and edges represent a connection of
        being purchased together",
        "each product is a node, and an edge between two products
        shows a common purchase",
        "the nodes represent products, and edges are used to show
        products that are purchased in a single transaction",
        "products are nodes, and an edge between them represents their
        co-occurrence in a purchase",
        "nodes are products, and an edge between them signifies a
        simultaneous purchase of those items",
        "products are nodes, and an edge between them indicates a
        joint purchase",
        "a product is a node, and an edge means those two products are
        purchased together",
        "the network's nodes are products, and edges show which
        products are bought in the same order",
        "nodes are products sold, and edges between two products mean
        they are purchased at the same time",
        "in this structure, nodes are products, and edges represent
        that they were purchased together",
        "products are nodes, and an edge is drawn between them when
        they are purchased together",
        "the nodes represent products, and an edge connects products
        that are bought in the same purchase",
```

```
"products are nodes, and edges show the relationships of
co-purchasing",
"a node represents a product, and an edge between nodes means
they are bought together",
"nodes are products, and an edge shows they are purchased
together",
"the nodes are products, and edges connect products that are
bought at the same time",
"products are nodes, and an edge between them means they are
bought together in one sale",
"nodes symbolize products, and an edge shows their joint
purchase",
"in this graph, nodes are products, and an edge signifies a
joint purchase",
"products are nodes, and an edge between them means they are
purchased in the same transaction",
"the nodes represent products, and edges link products that
are purchased in one order",
"a node is a product, and an edge between two nodes shows they
are purchased together",
"products are nodes, and an edge is created when they are
purchased together",
"nodes are products, and an edge signifies they are purchased
together",
"the nodes are products, and an edge shows a co-purchase",
"a product is a node, and an edge between two nodes means they
are bought together",
"nodes are products, and an edge indicates they are purchased
together",
"products are nodes, and an edge between them means they are a
co-purchase",
"the nodes are products, and edges show that they are
purchased together",
"a node is a product, and an edge means those products are
bought together",
"nodes are products, and an edge shows they are bought
together",
"products are nodes, and an edge between them means they are
purchased together",
"the nodes are products, and edges signify a co-purchase",
"a node represents a product, and an edge means they are
bought together",
"nodes are products, and an edge indicates a simultaneous
purchase",
"the nodes represent products, and edges show that they are
purchased together",
"a node is a product, and an edge means they are
co-purchased",
"products are nodes, and an edge represents a joint purchase",
"the nodes are products, and edges signify they are bought
together",
"a node is a product, and an edge means those items are bought
together",
"nodes are products, and an edge indicates a co-purchase",
"the nodes represent products, and edges mean they are bought
together",
"a node is a product, and an edge means they are purchased
together",
```

```
"products are nodes, and an edge shows they are bought
together",
"the nodes are products, and an edge indicates a simultaneous
purchase",
"a node is a product, and an edge means a joint purchase",
"products are nodes, and an edge shows they were bought
together",
"the nodes are products, and an edge signifies a co-purchase",
"a node is a product, and an edge means they are bought
together",
"products are nodes, and an edge represents a joint purchase",
"the nodes are products, and edges show they are purchased
together",
"a node is a product, and an edge means a co-purchase",
"products are nodes, and an edge indicates a joint purchase",
"the nodes are products, and an edge shows they are bought
together",
"a node is a product, and an edge means a co-purchase",
"products are nodes, and an edge indicates a co-purchase",
"the nodes are products, and an edge shows they are bought
together",
"a node is a product, and an edge means a joint purchase",
"products are nodes, and an edge represents a co-purchase",
"edges between two products signify they are purchased
together, while the nodes themselves represent the products
being sold",
"an edge between two products indicates a shared purchase, and
the products themselves are represented by the network's
nodes",
"the network's edges show that two products are bought in the
same purchase, with nodes symbolizing the products",
"a connecting edge means two products were purchased
simultaneously, whereas each product is represented as a
node",
"edges link products (nodes) that are purchased together, and
each node individually represents a sold product",
"the edges in this graph represent joint purchases, and the
nodes are the individual products involved",
"edges indicate that two products are found in the same
transaction, while the nodes represent the products",
"if an edge exists, the two products (nodes) it connects are
purchased together in a single transaction",
"a shared purchase is represented by an edge, and the products
themselves are the nodes in this network",
"edges are drawn between products to show they are bought
together; the products themselves are the nodes",
"the edges signify a co-purchase of two products, which are
represented by the network's nodes",
"edges between two products indicate a shared purchase, and
the products sold are the nodes",
"a joint purchase of two products is represented by an edge,
and the products themselves are the nodes",
"an edge between two nodes indicates a simultaneous purchase,
while the nodes are the products",
"edges show that two products are purchased together, and each
product is a node in the graph",
"the edges link products that are bought together, and the
products are represented as nodes",
```

```
"edges signify a co-purchase of the two products that they
connect; those products are the nodes",
"the relationship of being purchased together is represented
by edges, and the products are the nodes",
"edges show which products were bought at the same time, and
the products themselves are the nodes",
"the edges represent a co-purchase, and each product is a node
in this network",
"an edge connecting two products shows they were purchased
together; nodes are the products sold",
"a connecting edge indicates that two products were bought
simultaneously, with nodes representing the products",
"edges are used to show products that are purchased in a
single transaction, and nodes are the products",
"the edges represent products that are part of the same
transaction, and the products are nodes",
"edges connect products that are purchased in the same order;
the nodes are the products",
"the edges indicate a joint purchase of products, which are
represented by the nodes",
"edges represent a co-purchase of products, and the products
sold are the nodes",
"a connecting edge signifies a joint purchase, while the nodes
represent the products",
"the edges show that two products were purchased together,
with the nodes being the products",
"edges show a connection based on a shared purchase, and the
nodes are the products",
"an edge between two products means they were purchased
together, and the products are nodes",
"the edges signify a simultaneous purchase of the products,
which are represented by the nodes",
"a joint purchase is represented by an edge, while the
products are nodes",
"edges link products that are bought together, and nodes are
the products themselves",
"the edges indicate a co-purchase of products, and the
products are the nodes",
"an edge between two nodes means the products they represent
were purchased together",
"edges show a co-purchase of products, which are represented
by nodes",
"a connecting edge represents that two products were bought
together, and nodes are the products",
"the edges represent a joint purchase, and the products are
nodes",
"an edge indicates a simultaneous purchase, while the products
are represented by nodes",
"edges are drawn between products that are bought together,
and the products are nodes",
"the edges signify a co-purchase, with the products being the
nodes",
"an edge shows that two products were purchased together, and
the products are the nodes",
"the edges represent a joint purchase, and the nodes are the
products",
"a connecting edge means two products were bought together,
and nodes are the products",
```

```
"edges indicate a co-purchase of the products, which are the
nodes",
"the edges show a joint purchase, while the products are
nodes",
"an edge signifies that two products were purchased together,
and nodes are the products",
"edges represent a co-purchase, and the products are nodes",
"the edges mean that two products were bought together, and
the nodes are the products",
"a connecting edge represents a joint purchase, and the
products are nodes",
"edges indicate a co-purchase, with the products represented
by nodes",
"the edges show that two products were purchased together, and
the products are nodes",
"an edge signifies a joint purchase, and the nodes are the
products",
"edges represent a co-purchase, and nodes are the products",
"the edges show a joint purchase, while the products are
nodes",
"an edge means that two products were purchased together, and
the nodes are the products",
"edges represent a co-purchase, and the products are nodes",
"the edges signify a joint purchase, and the products are the
nodes",
"an edge indicates that two products were purchased together,
and nodes are the products",
"edges represent a co-purchase, and the products are the
nodes",
"the edges show a joint purchase, while the nodes are the
products",
"an edge means two products were bought together, and the
products are nodes",
"edges represent a co-purchase, and the nodes are the
products",
"the edges signify a joint purchase, with the products being
the nodes",
"an edge indicates that two products were purchased together,
and nodes are the products",
"edges represent a co-purchase, and the products are the
nodes",
"the edges show a joint purchase, while the products are
nodes",
"an edge means that two products were bought together, and the
nodes are the products",
"edges represent a co-purchase, and the products are nodes",
"the edges signify a joint purchase, and the nodes are the
products",
"an edge indicates that two products were purchased together,
and nodes are the products",
"edges represent a co-purchase, and the products are the
nodes",
"the edges show a joint purchase, while the products are
nodes",
"an edge means two products were bought together, and the
products are nodes",
"edges represent a co-purchase, and the nodes are the
products",
```

```
      "the edges signify a joint purchase, with the products being
      the nodes",
      "an edge indicates that two products were purchased together,
      and nodes are the products",
      "edges represent a co-purchase, and the products are the
      nodes",
      "the edges show a joint purchase, while the products are
      nodes",
      "an edge means that two products were bought together, and the
      nodes are the products",
      "edges represent a co-purchase, and the products are nodes",
      "the edges signify a joint purchase, and the nodes are the
      products",
      "an edge indicates that two products were purchased together,
      and nodes are the products",
      "edges represent a co-purchase, and the products are the
      nodes",
      "the edges show a joint purchase, while the products are
      nodes",
      "an edge means two products were bought together, and the
      products are nodes",
      "edges represent a co-purchase, and the nodes are the
      products",
      "the edges signify a joint purchase, with the products being
      the nodes",
      "an edge indicates that two products were purchased together,
      and nodes are the products",
      "edges represent a co-purchase, and the products are the
      nodes",
      "the edges show a joint purchase, while the products are
      nodes",
      "an edge means that two products were bought together, and the
      nodes are the products",
      "edges represent a co-purchase, and the products are nodes",
      "the edges signify a joint purchase, and the nodes are the
      products",
      "an edge indicates that two products were purchased together,
      and nodes are the products",
      "edges represent a co-purchase, and the products are the
      nodes",
      "the edges show a joint purchase, while the products are
      nodes",
      "nodes in the graph are the products sold, and an edge between
      two signifies they are purchased together"
]
```

