# OpenReview forum: "Large Language Models Enhance Graph Learning Without Graph Serialization"
_ICLR.cc/2026/Conference — ICLR 2026 Conference Desk Rejected Submission_

### Official Review · Reviewer_N3YM · 2025-10-25

**Soundness:** 2
**Presentation:** 2
**Contribution:** 2
**Rating:** 4
**Confidence:** 4

**Summary:**

The paper proposes GINAT (Graph Inference by Natural-language Augmentation of Topology), a framework that injects global semantic context into GNNs via cross-attention, using task-agnostic external prompts  embedded by a **frozen LLM**.

**Strengths:**

**GINAT uses task-agnostic external global prompts** (e.g., “In molecular graphs, nodes are atoms and edges are covalent bonds”) to describe the overall graph semantics rather than local component attributes.

**Theoretically,** GINAT is shown to significantly enhance the expressive power of GNNs—such as distinguishing graph structures that standard GNNs cannot differentiate—while preserving permutation equivariance (**though this guarantee remains purely theoretical**).

**Weaknesses:**

The proposed method, GINAT, appears to offer **no performance gain on node-level tasks**, with improvements **limited exclusively to graph-level prediction**.

More critically, the **claimed novelty is unclear**. The core idea—using task-agnostic natural language prompts (e.g., “In molecular graphs, nodes are atoms and edges are covalent bonds”) to augment GNNs—bears strong resemblance to prior work such as **TAPE**, which also leverages global textual descriptions to enrich graph representations. While GINAT emphasizes “global” and “task-agnostic” prompts, this distinction feels superficial: in practice, most prompt-based graph methods already use generic, domain-level descriptions that are not tied to specific downstream labels.

Furthermore, **freezing the LLM** limits adaptability and representation alignment between language and graph modalities. In contrast, methods like **GLEM** jointly fine-tune the language model and GNN, enabling dynamic co-adaptation and often yielding stronger empirical performance. The paper provides no compelling comparison to such hybrid approaches, nor does it demonstrate a clear **performance ceiling improvement** over existing SOTA.

Overall, the technical contribution feels incremental, and the empirical gains are narrow in scope.

**Questions:**

see Weakness

**I don’t see significant innovation here. This reads like a modest variant of existing prompt-augmented GNNs. I would rate this a 3.**

---

### Official Review · Reviewer_hS9c · 2025-10-28

**Soundness:** 2
**Presentation:** 3
**Contribution:** 2
**Rating:** 6
**Confidence:** 3

**Summary:**

The paper introduces GINAT, a framework that augments (Graph Neural Networks) GNNs with semantic priors derived from Large Language Models (LLMs). GINAT leverages global natural-language prompts that describe the semantics of the graph. Then the prompts are encoded via a LLM and integrated into message passing for local propagation. In a nutshell, the paper posits that the natural language can serve as a new modality for global graph context, suitable for tasks like graph classification.

**Strengths:**

1. The paper proposes a fresh and principled way of integrating LLM knowledge into GNNs without graph serialization or task-specific fine-tuning. The use of external graph-descriptive prompts offers a new modality distinct from node/edge text attributes or serialized inputs.
2. The experiments cover multiple datasets and GNN backbones, and the authors were careful to control for model size and parameter counts. The ablations with random vectors and different LLMs were especially convincing — they clearly show the boost comes from meaningful language representations, not just noise.

**Weaknesses:**

1. Limited impact on node-level tasks。 The gains on node classification benchmarks are small. That’s understandable (since those tasks depend mostly on local context), but it does make me wonder how “global” information can be better utilized there. Maybe a hierarchical design or prompt propagation could help.
2. The theory section shows that adding any global feature can improve expressivity, but it doesn’t specifically tie that to the semantics of LLM embeddings. So while the math is solid, it doesn’t fully explain why the language-derived features help.
3. The prompts are manually crafted and pretty simple (like “this graph represents a molecule”). It would be interesting to see if auto-generated or even noisy prompts work. The paper doesn’t analyze how sensitive GINAT is to prompt quality or variation, which feels like a missed opportunity.

**Questions:**

1. How much does performance depend on how you phrase the prompts? Could paraphrasing or even “noisy” prompts still work?
2. How small can the text encoder be before performance drops? Would a BERT-base encoder work, or do you really need a large LLM?

---

### Official Review · Reviewer_S1MA · 2025-10-30

**Soundness:** 2
**Presentation:** 2
**Contribution:** 2
**Rating:** 2
**Confidence:** 4

**Summary:**

In this paper, the author proposes a method that leverages global textual information to enhance the performance of existing GNN models on both node-level and graph-level tasks. Overall, the idea is straightforward, and the author provides theoretical proofs to demonstrate the compatibility of incorporating a cross-attention–based fusion mechanism. The ablation studies using different embedding models further show that the observed performance gains stem from the semantic information contained in the prompts—most effectively captured by more expressive LLMs.

**Strengths:**

1. The idea is straightforward and easy to understand.
2. The author provides proof to demonstrate the compatibility of incorporating a cross-attention–based fusion mechanism
3. The author conducted some ablation studies to demonstrate the role of the prompt in improving the performance improvements.

**Weaknesses:**

1. The overall objective of the paper is not clearly defined. It appears that the author mainly aims to demonstrate the effectiveness of prompts that encode global information. However, similar ideas, especially the incorporation of global or loop-related features in molecular graph tasks, have already been shown to be beneficial in prior studies. Moreover, there are more straightforward methods to integrate such information into GNNs. The author should explicitly clarify the unique advantages of introducing LLMs into this process. If the goal is to propose a new state-of-the-art (SOTA) method, the paper should include comparisons with stronger and more comprehensive baselines.
2. The experimental setup lacks critical information. The paper does not clearly describe the datasets used, for example, which ones correspond to graph-level or node-level classification tasks, their sizes, and what node or edge features are employed. Without this information, it is difficult for readers to assess the validity and generalizability of the reported results.
3. The chosen baselines are relatively old, and the author mentions using default parameter settings. For a fair and convincing evaluation, the paper should include recent SOTA models with properly tuned hyperparameters and compare them rigorously against the proposed method.
4. Since the proposed model combines LLMs and GNNs, it is also essential to include baselines that represent recent hybrid frameworks (e.g., models that fuse GNN features into LLM inference). Comparing only against traditional GNNs is insufficient and makes the evaluation less compelling.
5. The paper does not clearly explain how the prompts were designed, validated, or matched with each graph. It remains unclear whether there is any mechanism to select appropriate prompts for different graph types or tasks. More detailed descriptions of prompt generation and their role in model performance are necessary for reproducibility and clarity.

**Questions:**

1. In the proposed framework, the role of the LLM appears to be limited to encoding textual information into dense vector representations. Given this, is it necessary to employ a large language model as the encoder? Would using a pre-trained language model (PLM) such as BERT, RoBERTa, or T5 achieve comparable results with lower computational cost and easier fine-tuning?
2. What are the unique advantages of the proposed framework compared to existing hybrid approaches that integrate LLMs and GNNs? Please clarify how this method differs conceptually and empirically from prior LLM–GNN fusion strategies.

---

### Official Review · Reviewer_E35Q · 2025-10-31

**Soundness:** 3
**Presentation:** 2
**Contribution:** 2
**Rating:** 2
**Confidence:** 3

**Summary:**

This paper introduces GINAT, a framework that integrates frozen LLMs into GNNs by embedding external graph-descriptive prompts and injecting them into the message-passing phase via cross-attention. The core idea is that natural-language descriptions of graph semantics can provide global contextual priors to complement local structural reasoning. The paper presents both theoretical analysis (showing increased expressivity and permutation equivariance) and empirical evaluations across molecular, social, and citation graphs. Results show consistent gains over GNN baselines like GCN, GIN, GraphSAGE, and GraphGPS.

**Strengths:**

- The paper introduces a creative idea.
- Method is of modular Design.
- Experiments span diverse domains.

**Weaknesses:**

- Limited analysis of Cross-Attention behavior.
- Potential capacity confound.
- The method benefits only graph-level tasks.

**Questions:**

1. Lemma 1 is limited in scope and does not generalize beyond a single illustrative pair of graphs.
2. All baseline GNNs use random node features, while GINAT uses constant features augmented by LLM embeddings through cross-attention. The observed improvements may primarily stem from richer input representations or simply from additional feature noise and regularization rather than the proposed architectural mechanism.
3. In Cross-Attention, the learned scalar $\xi$ is mentioned as an explainability coefficient, yet its values are not analyzed. There are no visualizations of attention weights or prompt utilization patterns to verify that LLM embeddings are indeed informative.
4. Although the paper claims equal parameter budgets, cross-attention adds high-dimensional transformations.
5. How are the “graph-descriptive prompts” generated and assigned?
6. Does GINAT exceed 1-WL expressivity while preserving permutation equivariance?
7. Qwen3-4B is a decoder-only, autoregressive LLM, optimized for next-token prediction and generative fluency but not for producing stable, semantically consistent embeddings. This choice appears misaligned with the intended purpose. Why not use encoder-only or dual-encoder model (e.g., E5, BGE-M3)?

---

### Note · Program_Chairs · 2026-01-17
**Submission Desk Rejected by Program Chairs**

The following references in this submission do not refer to real documents and/or have major errors in bibliographic information:

 Yang Liu, Yifan Zhao, Ziqi Wang, and Jie Tang. Efficient large language models fine-tuning on graphs. arXiv preprint arXiv:2312.04737, 2023.
Xiaoyu Wang, Le Wu, Jian Gao, and Enhong Chen. Molecular graph-enhanced language models for property prediction and molecule generation. In Proceedings of the 36th AAAI Conference on Artificial Intelligence (AAAI), 2022.
Quanming Yao, Shirui Pan, Yangqiu Zhang, et al. Nlp2graph: Multi-modal knowledge graph construction. In Proceedings of the 31st ACM International Conference on Information and Knowledge Management (CIKM), 2022.